# Prototypical Hash Encoding for On-the-Fly Fine-Grained Category Discovery

**Haiyang Zheng**[1][*], **Nan Pu**[1][*], **Wenjing Li**[2][†], **Nicu Sebe**[1], **Zhun Zhong**[2][†]

[1]University of Trento    [2]Hefei University of Technology

## Abstract

In this paper, we study a practical yet challenging task, On-the-fly Category Discovery (OCD), aiming to online discover the newly-coming stream data that belong to both known and unknown classes, by leveraging only known category knowledge contained in labeled data. Previous OCD methods employ the hash-based technique to represent old/new categories by hash codes for instance-wise inference. However, directly mapping features into low-dimensional hash space not only inevitably damages the ability to distinguish classes and but also causes "high sensitivity" issue, especially for fine-grained classes, leading to inferior performance. To address these issues, we propose a novel Prototypical Hash Encoding (PHE) framework consisting of Category-aware Prototype Generation (CPG) and Discriminative Category Encoding (DCE) to mitigate the sensitivity of hash code while preserving rich discriminative information contained in high-dimension feature space, in a two-stage projection fashion. CPG enables the model to fully capture the intra-category diversity by representing each category with multiple prototypes. DCE boosts the discrimination ability of hash code with the guidance of the generated category prototypes and the constraint of minimum separation distance. By jointly optimizing CPG and DCE, we demonstrate that these two components are mutually beneficial towards an effective OCD. Extensive experiments show the significant superiority of our PHE over previous methods, *e.g.,* obtaining an improvement of +5.3% in ALL ACC averaged on all datasets. Moreover, due to the nature of the interpretable prototypes, we visually analyze the underlying mechanism of how PHE helps group certain samples into either known or unknown categories. Code is available at https://github.com/HaiyangZheng/PHE.

## 1 Introduction

While deep learning based machines have surpassed humans in visual recognition tasks [1, 2, 3, 4], their capability is often limited to providing closed-set answers, *e.g.*, category names. In contrast, humans possess the ability to recognize novel categories upon first observation without knowing their category names. To bridge this gap, Novel/Generalized Category Discovery (NCD/GCD) techniques [5, 6, 7, 8, 9, 10, 11] are proposed to transfer knowledge from known categories to distinguish unseen ones. However, current NCD/GCD methods operate under an offline inference paradigm where the category discovery is often implemented by applying clustering / unsupervised classification algorithms on a pre-collected batch of query data that needs to be discovered. This severely limits the practicability of NCD/GCD techniques in real-world applications, where the systems are expected to provide online feedback for every newly-coming instance.

To tackle this drawback, Du et al. introduce the On-the-fly Category Discovery (OCD) task [12], which removes the assumption of a predefined query set and requires instance feedback with stream data input. OCD poses two primary challenges: **1)** The requirement for real-time feedback is

---

[*]Equal contribution.

[†]Corresponding authors.

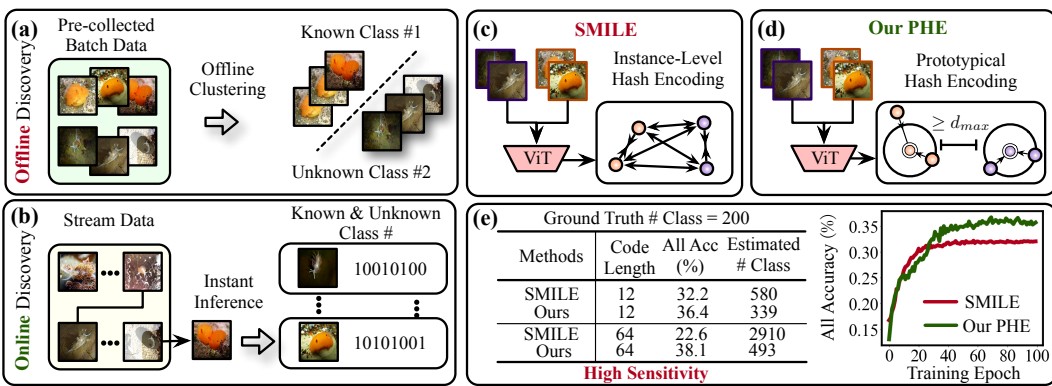

Figure 1: (a) Schema of Offline Category Discovery task (*e.g.*, NCD [5] and GCD [8]). (b) Schema of On-the-fly Category Discovery task [12], studied in this paper. (c) Previous work (*e.g.*, SMILE [12]) based on instance-level hash encoding. (d) Our PHE explores prototype-based hash encoding. (e) Performance comparison of PHE and SMILE and observation about "High Sensitivity".

incompatible with offline cluster-based methods [8, 9] (see Fig. 1 (a)). **2)** Due to uncertain open-world scenarios, the NCD/GCD methods (*e.g.*, classifier-based methods [13]), which assume the number of categories can be available/discovered as a prior, are ineffective for OCD. As a solution, Du et al. propose the SMILE architecture [12], which employs the sign of an image feature (i.e., hash code) as its category descriptor to identify the corresponding category for online discovery. As illustrated in Fig. 1 (c), SMILE directly maps image features into low-dimensional hash space with an instance-level contrastive objective and regard that one hash code uniquely represents a category. Given this, although SMILE can derive category descriptors, it suffers from a significant issue of "high sensitivity" for learned hash-form category descriptors and thus produces a significantly **inaccurate number of categories** as well as **unsatisfied performance**. As shown in Fig. 1 (e), this drawback of SMILE is even more severe for fine-grained categories due to the small inter-class variations and large intra-class variations between samples. For example, given two images of the same bird category but with different poses, they commonly share very similar overall features. However, under SMILE, minor variations in a single feature dimension can lead to opposite signs in their hash codes when the feature values are close to zero, classifying these two samples into distinct categories. Moreover, this issue becomes more pronounced as the feature dimensionality increases, as demonstrated in Table 5.

To address this issue, we propose a new two-stage Prototypical Hash Encoding (PHE) framework to improve the intra-class compactness and inter-class separation of category descriptors while alleviating information loss due to dimension reduction. PHE is composed of Category-aware Prototype Generation (CPG) and Discriminative Category Encoding (DCE). First, CPG utilizes prototype-based interpretable classification models to learn multiple prototypes for each category and yields sparse representations with a probabilistic masking strategy. Then, DCE explicitly maps the prototypes of each category to corresponding category hash centers. By imposing hash codes approaching their hash centers, OCD models are encouraged to produce more accurate hash codes. Meanwhile, we design a center separation loss to ensure that different category hash centers maintain at least a Hamming distance of $d_{max}$, where the $d_{max}$ is derived from Gilbert-Varshamov bound in coding theory, as illustrated in Fig. 1 (d). Finally, we design a tailor-made OCD model inference method based on a relaxed Hamming ball condition. Overall, the two-stage PHE framework enables OCD models to maintain the discriminabilities learned in high-dimension feature space and transfer them into low-dimensional encoding space, thereby enhancing accuracy for both seen and unseen categories. Extensive experiments conducted across multiple fine-grained benchmarks demonstrate that our approach substantially outperforms the SMILE architecture (see Fig. 1 (e)). Our contributions are summarized as follows:

- We propose a new PHE framework to explicitly generate prototypical hash centers, which is beneficial for improving the intra-class compactness and the inter-class separation of hash codes. Our PHE can effectively alleviate the "high sensitivity" issue of hash-based OCD methods.
- We design a tailor-made center separation loss to further improve the discriminability of hash centers by constraining a minimal separation distance derived from coding theory [14]. We also provide visual analyses to better understand the underlying mechanism of our PHE.
- Experiments on eight fine-grained datasets show that our method outperforms previous methods by a large margin, *i.e.*, +5.3% improvement averaged on all datasets for all class accuracy.

## 2 Related Works

**Novel Category Discovery** (NCD), initially introduced by DTC [5], aims to categorize unlabeled novel classes by transferring knowledge from labeled known classes. However, existing NCD methods [5, 6, 7] assume that all unlabeled data exclusively belong to novel classes. Generalized Category Discovery (GCD) is proposed in [8], allowing unlabeled data to be sampled from both novel and known classes. While existing NCD/GCD methods [6, 7, 8, 9, 13, 15, 16] have shown promising results, two key assumptions still impede their real-world application. Firstly, these models heavily rely on a predetermined query set (the unlabeled dataset) during training, limiting their ability to handle truly novel samples and hindering generalization. Secondly, the offline batch processing of the query set during inference makes these models impractical for online scenarios where data emerges continuously and the model requires instant feedback. To address these limitations, Du et al. introduced the On-the-Fly Category Discovery (OCD) [12], which removes the assumption of a predefined query set and requires instance feedback with stream data input. They proposed the SMILE method, which identifies the category of each instance by the signature of its representation (a hash-form code). The comparison among different settings is shown in Table 1.

In this paper, we address the "high sensitivity" issue associated with hash-based category descriptors in SMILE, particularly in fine-grained scenarios. We encode category prototypes into hash centers, ensuring maximal separation between these centers. Additionally, we employ Hamming balls centered on these hash centers to represent each category, effectively mitigating the "high sensitivity" issue.

Table 1: Comparison between different category discovery settings. $^*$ indicates that the number of new classes (Cls) is set as the ground-truth or previously estimated.

| Setting | Training Data | | | Test Data | | | |
|---|---|---|---|---|---|---|---|
| | Old Cls | New Cls | Require #Cls | Old Cls | New Cls | Require # Cls | Online |
| NCD | ✓ | ✓ | Old+New$^*$ | ✗ | ✓ | Old+New$^*$ | ✗ |
| GCD | ✓ | ✓ | Old+New$^*$ | ✓ | ✓ | Old+New$^*$ | ✗ |
| OCD | ✓ | ✗ | Old | ✓ | ✓ | Old | ✓ |

**Deep Hashing** is a popular method for large-scale image retrieval. It uses deep neural networks to learn a hash function that converts samples into fixed-length binary codes, ensuring similar samples share similar codes. Early methods, such as HashNet [17], DPSH [18], and DSH [19], optimize hash functions based on pairwise similarities or triplet-based similarities. Both approaches suffer from low training efficiency and insufficient data distribution coverage. By defining points as hash centers that are sufficiently spaced apart, methods like DPN [20], OrthoHash [21], and CSQ [22] largely enhance training efficiency and retrieval accuracy. However, in the worst case, hash centers derived from these methods can be arbitrarily close. To address this, Wang et al. [23] introduce the Gilbert-Varshamov bound from coding theory to ensure a large minimal distance between hash centers.

Drawing inspiration from deep hash methods, we employ a hash center-based approach for category encoding. Unlike deep hashing methods used in image retrieval tasks [22, 23], which rely on predefined center points, our method generates category-specific hash centers directly from category prototypes. This adaptation is particularly suitable for the category discovery task. Furthermore, we utilize Hamming balls centered on these hash centers to represent categories, effectively mitigating the "high sensitivity" issue associated with using hash-form descriptors in category discovery tasks.

**Prototype-based Interpretable Models.** Chen et al. [24] introduce the Prototypical Part Network (ProtoPNet), a model designed for Interpretable classification. ProtoPNet features a set number of prototypical parts per class, enabling clear decision-making process. In addition, ProtoPNet offers a post-hoc analysis, in which it explains decisions for individual images by displaying all prototypes alongside their weighted similarity scores. This method employs multiple prototypes to represent a category, effectively distinguishing subtle differences between categories and demonstrating strong performance in fine-grained classification. Numerous subsequent studies have adapted ProtoPNet for various applications such as medical image processing and explanatory debugging, among others [25, 26, 27, 28, 29]. Later, ProtoPFormer [30] further integrates the Vision Transformer as a backbone, utilizing both global and part prototypes for interpretable image classification.

In this paper, we utilize prototype-based models for representation learning and prototype acquisition. Unlike existing methods predominantly tailored for closed-set classification, our approach extends to the generation of discriminative hash codes from the learned category prototypes. This extension allows for broader applicability in handling new and unseen categories, *i.e.*, open-set scenarios.

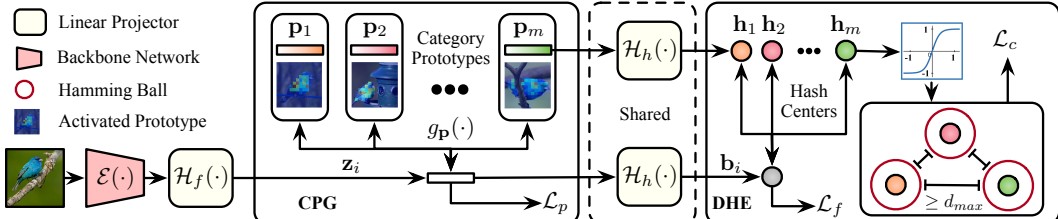

Figure 2: Our PHE framework is composed of the CPG and DHC modules. First, CPG generates category-specific prototypes and prototype-guided instance representations. Then, DHC encodes the generated prototypes as hash centers to encourage the model to learn discriminative instance hash codes. Finally, depending on the Hamming distance between instance hash codes and hash centers, we can obtain instant feedback and online group instances into both known and unknown categories.

## 3 Prototypical Hash Encoding

**Problem Setup.** The setting of OCD is defined as follows. We are provided with a support set, denoted as $\mathcal{D}_S = \{(\mathbf{x}_i, y_i^s)\}_{i=1}^M \subseteq \mathcal{X} \times \mathcal{Y}_S$ for training, and a query set, denoted as $\mathcal{D}_Q = \{(\mathbf{x}_i, y_i^q)\}_{i=1}^N \subseteq \mathcal{X} \times \mathcal{Y}_Q$ for testing. Here, $M$ and $N$ represent the number of samples in $\mathcal{D}_S$ and $\mathcal{D}_Q$, respectively. $\mathcal{Y}_S$ and $\mathcal{Y}_Q$ are the label spaces for the support set and query set, respectively, where $\mathcal{Y}_S \subseteq \mathcal{Y}_Q$. We define classes in $\mathcal{Y}_S$ as known/old classes and classes in $\mathcal{Y}_Q/\mathcal{Y}_S$ as unknown/new classes. Only the support set $\mathcal{D}_S$ is used for model training. During testing, $\mathcal{D}_Q$ includes samples from both known and unknown categories, which are inferred one by one, allowing for instant/online feedback.

**Framework Overview**. To achieve accurate and online category discovery, we design a Prototypical Hash Encoding (PHE) framework, which mainly consists of a Category-aware Prototype Generation (CPG) module and a Discriminative Hash Encoding (DHE) module, as illustrated in Fig. 2. CPG aims at modeling diverse intra-category information and generating category-specific prototypes for representing fine-grained categories. DHE leverages generated prototypical hash centers to further facilitate discriminative hash code generation. Finally, based on a theoretically derived bounding of the Hamming ball, we can determine the under-discovered category of instance and acquire instant feedback.

### 3.1 Category-aware Prototype Generation

SMILE [12] directly utilizes instance-level supervised contrastive learning on low-dimensional hash features for simultaneous representation learning and category encoding. This approach may result in inadequate representation learning, as low-dimensional features struggle to capture complex data structures and patterns, especially in challenging fine-grained scenarios (see Table 10).

To solve this issue, we choose to perform representation learning upon the high-dimensional features instead of the low-dimensional hash features. Specifically, given a batch of input images, $\mathbf{X}$, the image features are denoted as $\mathbf{Z} = \mathcal{H}_f(\mathcal{E}(\mathbf{X})) \in \mathbb{R}^{N \times \hat{L}}$, where $\mathcal{E}$ represents the backbone, $\mathcal{H}_f$ denotes a linear head, and $\hat{L}$ represents the feature dimension. We use a prototype layer $g_\mathbf{p}$ to transform $\mathbf{Z}$ into a similarity score vector $\mathbf{s} \in \mathbb{R}^m$, with $g_\mathbf{p}$ containing $m$ learnable prototypes $\{\mathbf{p}_1, \mathbf{p}_2, \ldots, \mathbf{p}_m\}$. In the design of the prototype layer, CNN-based methods [24] typically utilize the maximum pooled value from the similarity map, which is calculated between the feature map and the prototype, as the similarity score. In this paper, to align with the Vision Transformer (ViT) [4] backbone commonly-used in OCD, we propose to employ its *cls token* to compute the prototype similarity score, inspired by ProtopFormer [30]. The similarity score $\mathbf{s}_{ij}$ for the $i$-th sample to the $j$-th prototype is calculated as follows:

$$\mathbf{s}_{i \to j} = g_{\mathbf{p}_j}(\mathbf{z}_i) = \log\left(\frac{\|\mathbf{z}_i - \mathbf{p}_j\|_2^2 + 1}{\|\mathbf{z}_i - \mathbf{p}_j\|_2^2 + \epsilon}\right), \quad (1)$$

where $\epsilon$ is a small constant for numerical stability.

**Probabilistic Masking Strategy.** To encourage models to fully capture intra-category diversity, we assign $k$ prototypes equally to each category, thus $m = k * |\mathcal{Y}_S|$. Given the prototype similarity score to all prototypes in the prototype layer of $i$-th image $\mathbf{s}_i = [\mathbf{s}_{i \to 1}, \mathbf{s}_{i \to 2}, ..., \mathbf{s}_{i \to m}]$, a fully connected layer (FC) is employed for supervised classification. To further discretize the prototypes,

avoid redundancy, and unleash the full potential of the prototypes, we utilize a probabilistic masking strategy. This strategy masks some units of the similarity score according to probability, thereby disabling the corresponding prototypes and leaving the remaining prototypes activated during training. Specifically, for each unit of similarity score $\mathbf{s}_i$, we mask it with a probability following the Bernoulli distribution $\mathcal{B}(\theta)$, with $\theta$ empirically set to 0.1. We use the following loss function to learn the image representations and prototypes:

$$\mathcal{L}_p = \frac{1}{|B|} \sum_{i \in B} \ell(\boldsymbol{y}_i, FC(\mathcal{B}(\theta) \cdot \mathbf{s}_i)), \tag{2}$$

where $B$ indicates the mini-batch of support set, $\ell$ is the traditional cross-entropy loss and $\boldsymbol{y}_i$ is the ground truth of image $\mathbf{x}_i$.

**Discussion.** The benefits of our CPG module are threefold: i) *Fine-Grained Category Distinction:* Fine-grained categories often exhibit only subtle differences. Our CPG module allows for representing a category with multiple prototypes, which can effectively capture and model both intra-class similarities and inter-class variances. ii) *Category-specific Hash Center Generation:* Based on the generated category prototypes of CPG, we further map them into low-dimensional hash features to serve as hash centers for each category. This helps in maintaining the distinctiveness of each category in the encoding space. iii) *A New perspective for Classification Analysis:* Beyond merely using hash encoding for category decisions, we can introduce a new prototype perspective for classification analysis. CPG enables a visualizable reasoning process for classifying unseen categories, providing a more intuitive and interpretable method for understanding category distinctions.

## 3.2 Discriminative Hash Encoding

**Category Encoding Learning.** Given the set of learned category prototypes $P_{c_i}$ for category $c_i$ in CPG, we map the mean vector of each category's prototypes $\sum P_{c_i}/k$ to its hash center, denoted as $\mathbf{h}_i = \mathcal{H}_h(\sum P_{c_i}/k) \in \mathbb{R}^L$, where $\mathcal{H}_h$ represents a linear head and $L$ represent the feature dimension. The image feature $\mathbf{z}_i$ is mapped to a hash feature $\mathbf{b}_i$, where $\mathbf{b}_i = \mathcal{H}_h(\mathbf{z}_i)$. To ensure that image representations of the same category as closely as possible share the same category descriptor, we constrain each hash feature to be close to its corresponding category hash center and distant from other hash centers. We employ the following loss to optimize the hash features of the images:

$$\mathcal{L}_f = \frac{1}{|B|} \sum_{i \in B} \ell(\boldsymbol{y}_i, sim(\mathbf{b}_i, \mathbf{h})), \tag{3}$$

where $sim(\mathbf{b}_i, \mathbf{h})$ represents a pair-wise similarity vector consisting of the cosine similarities between the hash feature of image $\mathbf{x}_i$ and all hash centers.

**Hash Centers Optimization.** Due to the subtle differences between fine-grained categories, different category hash centers may become closely similar or even share identical hash codes. This hinders separability between fine-grained categories and leads to incorrect classification results. Therefore, we aim to maximize the differences between hash centers for enhancing inter-class separation.

Given the hash center $\mathbf{h}_i$ of category $c_i$, we denote its hash code as $\hat{\mathbf{h}}_i$, where $\hat{\mathbf{h}}_i = sign(\mathbf{h}_i)$ and $sign(\cdot)$ equals 1/-1 for positive/negative values. Since the sign function is non-differentiable, we use a smoothed version of the sign function for back-propagation during training, defined as:

$$sign^*(a) \approx \frac{e^{a \times \tau} - e^{-a \times \tau}}{e^{a \times \tau} + e^{-a \times \tau}}, \tag{4}$$

where $\tau$ is a hyper-parameter that controls the smoothness of the sign function. Larger values of $\tau$ make the function more closely approximate the true sign function. In this paper, we set $\tau = 3$. Consequently, $\hat{\mathbf{h}}_i = sign^*(\mathbf{h}_i)$. The difference between the $i$-th and $j$-th hash centers can be evaluated by the Hamming distance of their hash codes: $||\hat{\mathbf{h}}_i - \hat{\mathbf{h}}_j||_H = \frac{L - \hat{\mathbf{h}}_i \cdot \hat{\mathbf{h}}_j}{2}$. Although we aim to maximize the differences between hash centers, we cannot simply increase the Hamming distance between all hash centers, as this can lead to model non-convergence. Since the encoding space is fixed, excessively distancing one hash code can inadvertently bring it closer to another.

We design a center separation loss, $\mathcal{L}_{sep}$, to ensure that the Hamming distance between any two hash centers is greater than or equal to $d$, denoted as $||\hat{\mathbf{h}}_i - \hat{\mathbf{h}}_j||_H \geq d$. The center separation loss is defined as:

$$\mathcal{L}_{sep} = \sum_i \sum_j \max(0, d - ||\hat{\mathbf{h}}_i - \hat{\mathbf{h}}_j||_H), \tag{5}$$

**Algorithm 1:** Pseudocode for Inference of Our Method

---

**Input:** Test data (query set) $\mathcal{D}_Q$ contains both known and unknown categories, trained PHE model contains { backbone $\mathcal{E}(\cdot)$, linear projector $\mathcal{H}_f(\cdot)$, prototype layer $g_{\mathbf{P}}(\cdot)$ and linear projector $\mathcal{H}_h(\cdot)$}, $d_{max}$.

1   Compute hash centers $\hat{\mathbf{h}}$ for known categories ;            `// Hash Centers`

2   Compute Hamming ball radius $\mathcal{R} = \max(\lfloor \frac{d_{max}}{2} \rfloor, 1)$;

3   Build a category list $\mathcal{C}_H$, add $\hat{\mathbf{h}}$ to $\mathcal{C}_H$;

4   **for** *each image* $\mathbf{x}_i \in \mathcal{D}_Q$ **do**

5      Compute hash code $\hat{\mathbf{b}}_i = sign(\mathcal{H}_h(\mathcal{H}_f(\mathcal{E}(\mathbf{x}_i))))$ ;          `// Category Descriptor`

6      **for** *hash center* $\hat{\mathbf{h}}_j \in \mathcal{C}_H$ **do**

7         Compute Hamming distance $||\hat{\mathbf{b}}_i - \hat{\mathbf{h}}_j||_H$ ;

8         **if** $||\hat{\mathbf{b}}_i - \hat{\mathbf{h}}_j||_H \leq \mathcal{R}$ **then**

9            **Output** pred $\hat{y}_i = \mathcal{C}_H.\text{index}(\hat{\mathbf{h}}_j)$ ;        `// Close to a known category`

10      /* **Different from existing categories** */ ;

11      Add $\hat{\mathbf{b}}_i$ to $\mathcal{C}_H$ ;                        `// Create a new category`

12      **Output** pred $\hat{y}_i = |\mathcal{C}_H|$ ;

---

indicating that we only optimize the pairs of hash centers whose Hamming distance is less than $d$. However, determining the appropriate value of $d$ is challenging. If $d$ is too large, it can lead to model non-convergence; if it is too small, it can cause inadequate separation between hash centers. We choose a maximum $d_{max}$ according to the Gilbert-Varshamum bound [14] in coding theory, as stated in Lemma 3.1.

**Lemma 3.1** *For binary symbols, there exists a set of hash codes of length $L$, $\{-1,1\}^L$, with a minimum Hamming distance $d$, and a number of hash codes $\mathcal{Q}$ that satisfies the following inequality:*

$$\mathcal{Q} \geq \frac{2^L}{\sum_{i=0}^{d-1} \binom{L}{i}}. \tag{6}$$

Therefore, given a number of hash codes equal to $|\mathcal{Y}_S|$, the maximum $d_{max}$ can be determined as:

$$\begin{cases} |\mathcal{Y}_S| \geq \frac{2^L}{\sum_{i=0}^{d_{max}-1} \binom{L}{i}}, \\ |\mathcal{Y}_S| \leq \frac{2^L}{\sum_{i=0}^{d_{max}-2} \binom{L}{i}}. \end{cases} \tag{7}$$

We apply this upper bound $d_{max}$ to $\mathcal{L}_{sep}$. Considering that we use a smoothed version of the sign function, where computed values do not equate to precisely $\pm 1$, we impose a quantization loss to constrain the values of hash codes to be close to $\pm 1$:

$$\mathcal{L}_q = \sum_i (1 - |\hat{\mathbf{h}}_i|). \tag{8}$$

The optimization loss $\mathcal{L}_c$ for hash centers is defined as a combination of the center separation loss $\mathcal{L}_{sep}$ and the quantization loss $\mathcal{L}_q$:

$$\mathcal{L}_c = \mathcal{L}_{sep} + \mathcal{L}_q. \tag{9}$$

### 3.3   Training and Inference

**Model Training.** During the model training process, the total loss is formulated as follows:

$$\mathcal{L} = \mathcal{L}_p + \alpha * \mathcal{L}_c + \beta * \mathcal{L}_f, \tag{10}$$

where $\alpha$ and $\beta$ control the importance of center optimization and hash encoding, respectively.

**Hamming Ball Based Model Inference.** During on-the-fly testing, given an input image $x_i$ in the query set $D_Q$, we use $\hat{\mathbf{b}}_i = sign(\mathcal{H}_h(\mathcal{H}_f(\mathcal{E}(\mathbf{x}_i))))$ as its category descriptor. Due to the introduction of the center separation loss, the Hamming distance between any two hash centers is not less than $d_{max}$. We consider a Hamming ball centered on the hash centers with a radius of $\max(\lfloor \frac{d_{max}}{2} \rfloor, 1)$ to represent a category. Specifically, during inference, if the Hamming distance between $\hat{\mathbf{b}}_i$ and any existing hash center is less than or equal to $\max(\lfloor \frac{d_{max}}{2} \rfloor, 1)$, we classify the image as belonging to the corresponding category of that hash center. Otherwise, the image is used to establish a new hash center and category. The pseudo-code for the inference is provided in Algorithm 1.

# 4 Experiment

## 4.1 Experiment Setup

**Datasets.** We have conducted experiments on eight fine-grained datasets, including CUB-200 [31], Stanford Cars [32], Oxford-IIIT Pet [33], Food-101 [34], and four super-categories from the more challenging dataset, iNaturalist [35], including Fungi, Arachnida, Animalia, and Mollusca. Following the setup in OCD [12], the categories of each dataset are split into subsets of seen and unseen categories. Specifically, 50% of the samples from the seen categories are used to form the labeled set $\mathcal{D}_S$ for training, while the remainder forms the unlabeled set $\mathcal{D}_Q$ for on-the-fly testing. Detailed information about the datasets used is provided in the Appendix. A.1.

**Evaluation Metrics.** We follow Du et al. [12] and adopt clustering accuracy as an evaluation protocol, formulated as $ACC = \frac{1}{|\mathcal{D}_Q|} \sum_{i=1}^{|\mathcal{D}_Q|} \mathbb{I}(y_i = C(\overline{y}_i))$, where $\overline{y}_i$ represents the predicted labels and $y_i$ denotes the ground truth. The function $C$ denotes the optimal permutation that aligns predicted cluster assignments with the actual class labels.

**Implementation Details.** For a fair comparison, we follow OCD [12] and use the DINO-pretrained ViT-B-16 [36] as the backbone. During training, only the final block of ViT-B-16 is fine-tuned. In our approach, the Projector $\mathcal{H}_f(\cdot)$ is a single linear layer with an output dimension set to $\hat{L} = 768$, meaning the feature dimension and prototype dimension are equal to 768. Each category has $k = 10$ prototypes. The FC layer in Eq. 2 is non-trainable, which uses positive weights 1 for prototypes from the same category and negative weights -0.5 for prototypes from different categories. The Projector $\mathcal{H}_h(\cdot)$ consists of three linear layers with an output dimension set to $L = 12$, which produces $2^{12} = 4096$ binary category encodings. By default, we follow OCD to set this dimension for fair comparison. Additional experiments with varying $L$ are reported in Sec. 4.4. The estimated values of $d_{max}$ in center separation loss $\mathcal{L}_{sep}$ can be found in Appendix. A.2. The ratio $\alpha$ and $\beta$ in the total loss are set to 0.1 and 3, respectively, for all datasets. More details and the pseudo-code for the training process can be found in the Appendix. A.2.

**Compared Methods.** Given that OCD is a relatively new task requiring instantaneous inference, traditional baselines from NCD and GCD are unsuitable for this setting. Consequently, we compared with the **SMILE** [12] along with three strong competitors in [12]: i) **Sequential Leader Clustering (SLC)** [37]: A classical clustering technique suitable for sequential data. ii) **Ranking Statistics (RankStat)** [38]: RankStat utilizes the top-3 indices of feature embeddings as category descriptors. iii) **Winner-take-all (WTA)** [39]: WTA employs indices of maximum values within feature groups as category descriptors. These three strong baselines are set following SMILE [12], and detailed implementation can be found in the Appendix. A.3.

## 4.2 Comparison with State of the Art

We conduct comparison experiments with the aforementioned competitors across eight datasets. The experimental results are reported in Table 2. It is evident that the proposed PHE outperforms all state-of-the-art competitors across nearly all metrics. In particular, when compared with three strong baselines—SLC, RankStat, and WTA—our method demonstrates significant improvements. Additionally, our method surpasses the top competitor, SMILE, by an average of 5.4% in All classes accuracy on four common fine-grained datasets, and by an average of 5.1% in All classes accuracy on the four challenging datasets from the iNaturalist dataset. We achieve an average improvement of 11.3% on Old classes across the eight datasets compared to SMILE, demonstrating the effectiveness of our method in alleviating the "high sensitivity" issue of hash-based OCD methods. Importantly, our method consistently improves accuracy for unseen/new classes compared to SMILE. For instance, we achieve a 4.1% improvement on CUB-200 and a 4.4% improvement on the Oxford Pets dataset for new classes. This demonstrates that our method exhibits stronger generalization capabilities to new classes compared to SMILE. This advantage becomes more pronounced as the dimensionality $L$ of hash features increases, as discussed in Sec. 4.4.

## 4.3 Ablation Study

**Components Ablation.** We report an ablation analysis of the proposed components in our PHE on CUB dataset and Stanford Cars dataset, as shown in Table 3. "Without $\mathcal{L}_p$" refers to the removal of the

Table 2: Comparison with the State of the Art methods. The best results are marked in **bold**, and the second best results are marked by underline.

| Method | CUB | | | Stanford Cars | | | Oxford Pets | | | Food101 | | | Average | | |
|---|---|---|---|---|---|---|---|---|---|---|---|---|---|---|---|
| | All | Old | New | All | Old | New | All | Old | New | All | Old | New | All | Old | New |
| SLC [37] | 31.3 | 48.5 | 22.7 | 24.0 | 45.8 | 13.6 | 35.5 | 41.3 | 33.1 | 20.9 | 48.6 | 6.8 | 27.9 | 46.1 | 19.1 |
| RankStat [38] | 27.6 | 46.2 | 18.3 | 18.6 | 36.9 | 9.7 | 33.2 | 42.3 | 28.4 | 22.3 | 50.7 | 7.8 | 25.4 | 44.0 | 16.1 |
| WTA [39] | 26.5 | 45.0 | 17.3 | 20.0 | 38.8 | 10.6 | 35.2 | 46.3 | 29.3 | 18.2 | 40.5 | 6.1 | 25.0 | 42.7 | 15.8 |
| SMILE [12] | 32.2 | 50.9 | 22.9 | 26.2 | 46.7 | 16.3 | 41.2 | 42.1 | 40.7 | 24.0 | 54.6 | 8.4 | 30.9 | 48.6 | 22.1 |
| PHE (Ours) | **36.4** | **55.8** | **27.0** | **31.3** | **61.9** | **16.8** | **48.3** | **53.8** | **45.4** | **29.1** | **64.7** | **11.1** | **36.3** | **59.1** | **25.1** |

| Method | Fungi | | | Arachnida | | | Animalia | | | Mollusca | | | Average | | |
|---|---|---|---|---|---|---|---|---|---|---|---|---|---|---|---|
| | All | Old | New | All | Old | New | All | Old | New | All | Old | New | All | Old | New |
| SLC [37] | 27.7 | 60.0 | 13.4 | 25.4 | 44.6 | 11.4 | 32.4 | **61.9** | 19.3 | 31.1 | 59.8 | 15.0 | 29.2 | 56.6 | 14.8 |
| RankStat [38] | 23.8 | 50.5 | 12.0 | 26.6 | 51.0 | 10.0 | 31.4 | 54.9 | 21.6 | 29.3 | 55.2 | 15.5 | 27.8 | 52.9 | 14.8 |
| WTA [39] | 27.5 | 65.6 | 12.0 | 28.1 | 55.5 | 10.9 | 33.4 | 59.8 | 22.4 | 30.3 | 55.4 | 17.0 | 29.8 | 59.1 | 15.6 |
| SMILE [12] | 29.3 | 64.6 | 13.6 | 29.9 | 57.9 | 12.2 | 35.9 | 49.4 | 30.3 | 33.3 | 44.5 | **27.2** | 32.1 | 54.1 | 20.8 |
| PHE (Ours) | **31.4** | **67.9** | **15.2** | **37.0** | **75.7** | **12.6** | **40.3** | 55.7 | **31.8** | **39.9** | **65.0** | 26.5 | **37.2** | **66.1** | **21.5** |

Table 3: Ablation study on training components. The best results are marked in **bold**.

| $\mathcal{L}_p$ | $\mathcal{L}_c$ | $\mathcal{L}_f$ | CUB | | | SCars | | |
|---|---|---|---|---|---|---|---|---|
| | | | All | Old | New | All | Old | New |
| | ✓ | ✓ | 34.9 | 53.0 | 25.8 | 28.9 | 58.4 | 14.6 |
| ✓ | | ✓ | 32.0 | 43.4 | 26.4 | 24.1 | 40.2 | 16.3 |
| ✓ | ✓ | | 34.1 | 54.3 | 24.0 | 26.0 | 52.6 | 13.1 |
| ✓ | ✓ | ✓ | **36.4** | **55.8** | **27.0** | **31.3** | **61.9** | **16.8** |

Table 4: Ablation study on training strategy. The best results are marked in **bold**.

| Methods | CUB | | | SCars | | |
|---|---|---|---|---|---|---|
| | All | Old | New | All | Old | New |
| Fixed-**h** | 35.4 | 54.7 | 25.8 | 30.0 | 61.5 | 14.8 |
| Linear Cls | 35.6 | **57.8** | 24.6 | 29.4 | 63.6 | 13.0 |
| Supcon Cls | 35.3 | 57.5 | 24.2 | 29.5 | **64.1** | 12.8 |
| Ours | **36.4** | 55.8 | **27.0** | **31.3** | 61.9 | **16.8** |

representation learning in CPG, where we use randomly initialized vectors which are further mapped to hash centers. This configuration results in an average reduction of 1.95% in All classes accuracy across the two datasets. This variant of PHE shares the same architecture as SMILE, achieving clear decline on both datasets, demonstrating the effectiveness of our hash center-based category encoding method. It is noteworthy that removing $\mathcal{L}_c$ causes a significant performance reduction, especially in seen categories—for instance, a 12.4% decrease in CUB. This decline is due to the lack of $\mathcal{L}_c$, which results in insufficient separation of hash centers, causing multiple old classes to share the same hash encoding and thereby being classified as the same. Additionally, removing $\mathcal{L}_p$ also leads to performance degradation. This is because the hash encodings of features are not sufficiently close to the hash centers, resulting in inadequate learning of hash features.

**Strategy Ablation.** In Table 4, we evaluate the training strategies of our method. "Fixed-**h**" refers to the use of handcrafted hash points that satisfy Eq. 7. Although this design maintains separation between hash centers, it may alter the relationships between categories learned in CPG, leading to sub-optimal outcomes. "Linear Cls." and "Supcon Cls." represent the use of simple linear classification and supervised contrastive learning classification methods for representation learning, respectively, where "Fixed-**h**" is also applied due to the absence of category prototypes. Although these variants perform well on seen categories, their generalization capabilities are inferior to our full prototype-based method. This demonstrates the importance of integrating prototype learning to enhance generalization across unseen categories.

## 4.4 Evaluation

**Evaluation on Hash Code Length.** The hash code length $L$ is crucial for category inference as it directly determines the size of the prediction space, which equals to $2^L$. A larger $L$ value results in a greater number of category encodings, making the "high sensitivity" issue more severe. We evaluate our method and SMILE with different hash code lengths in Table 5. When small changes occur to $L$ (from 12 to 16), SMILE demonstrates stable results. However, with $L = 32$, SMILE experiences a decrease in average all-classes accuracy by 5.1% on two datasets, and with $L = 64$, the decrease is by 10.2% averaged on the two datasets. In contrast, our method, which employs hash center-based

Table 5: Results with different hash code length $L$. The best results are marked in **bold** for each $L$.

| $L$ | Methods | CUB#200 | | | Estimated | SCars#196 | | | Estimated |
| --- | --- | --- | --- | --- | --- | --- | --- | --- | --- |
| | | All | Old | New | #Class | All | Old | New | #Class |
| 16bit | SMILE | 31.9 | 52.7 | 21.5 | 924 | 27.5 | 52.5 | 15.4 | 896 |
| | Ours | **37.6** | **57.4** | **27.6** | **318** | **31.8** | **65.4** | **15.6** | **709** |
| 32bit | SMILE | 27.3 | 52.0 | 14.97 | 2146 | 21.9 | 46.8 | 9.9 | 2953 |
| | Ours | **38.5** | **59.9** | **27.8** | **474** | **31.5** | **64.0** | **15.8** | **762** |
| 64bit | SMILE | 22.6 | 45.3 | 11.2 | 2910 | 16.5 | 38.2 | 6.1 | 4788 |
| | Ours | **38.1** | **60.1** | **27.2** | **493** | **32.1** | **66.9** | **15.3** | **917** |

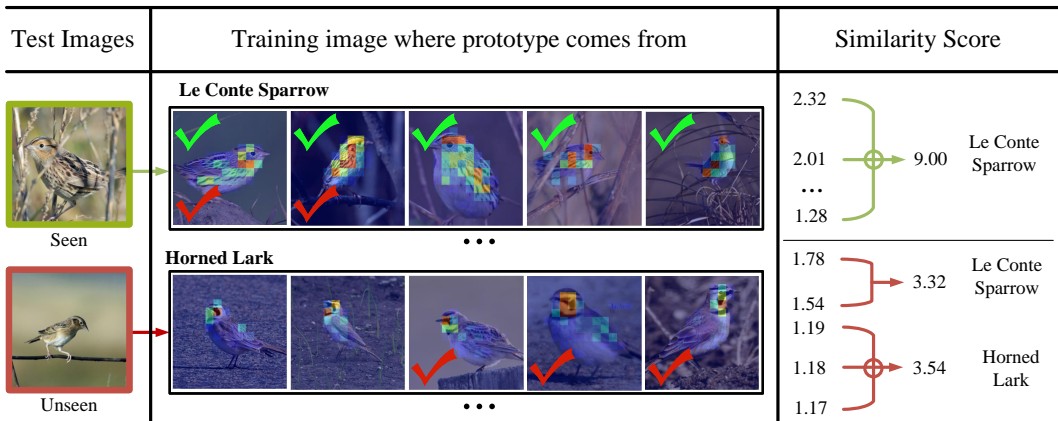

Figure 3: Case Study: Why is a Grasshopper Sparrow classified as a new category?

optimization and a Hamming ball-based inference process, effectively mitigates the "high sensitivity" issue and maintains stable performance as $L$ increases. Furthermore, when $L$ is increased from 16 to 64, the estimated number of classes by SMILE increased by 1986 on the CUB dataset and 3892 on the Stanford Cars dataset, underscoring the impact of "high sensitivity" of hash-form category descriptors on accuracy. Conversely, our method exhibits remarkable stability.

**Visualization Analysis.** We use an images from the support set whose latent representations is most similar to $\mathbf{p}_j$ as the visualization for $\mathbf{p}_j$. During on-the-fly inference, the hash code functions as the category descriptor. Additionally, the learned prototypes allow us to visually analyze why the model categorizes certain samples into known or unknown categories. Fig. 3 illustrates this reasoning process from the prototype perspective. On the left, an image labeled in green depicts a Le Conte Sparrow, a category recognized during training. Under it, a red-labeled image represents a Grasshopper Sparrow, a new category. In the center, we display the visualization of the prototypes, showing only five per class. The top five prototypes most similar to the green-labeled image are exclusively associated with the Le Conte Sparrow category, while those closest to the red-labeled, unseen image span two different categories. The Grasshopper Sparrow exhibits significant body stripe similarities to the Le Conte Sparrow and shares head similarities with the Horned Lark. Upon calculating similarities with the top five activated prototypes, the similarity score for the green-labeled image to the Le Conte Sparrow category is 9.0, whereas the unseen Grasshopper Sparrow achieves a similarity of 3.32 with the Le Conte Sparrow and 3.54 with the Horned Lark. This reasoning process is similar to how humans cognitively recognize new species. Given a new species, we use the characteristics of known categories to describe the unknown new category. Consequently, an entity that exhibits high similarity to multiple known categories, rather than only one known category, is likely to be classified as a previously unseen species.

**Hyper-parameters Analysis.** 1) The impact of the ratios $\alpha$ and $\beta$ in Eq. 10 is illustrated in Fig. 4. We use All classes accuracy as the evaluation metric. $\alpha$ and $\beta$ control the relative importance of $\mathcal{L}_c$ and $\mathcal{L}_f$ during the training process, respectively. A lower value of $\alpha$ is found to be preferable, as higher values can lead to excessive changes in hash centers, thereby affecting training stability.

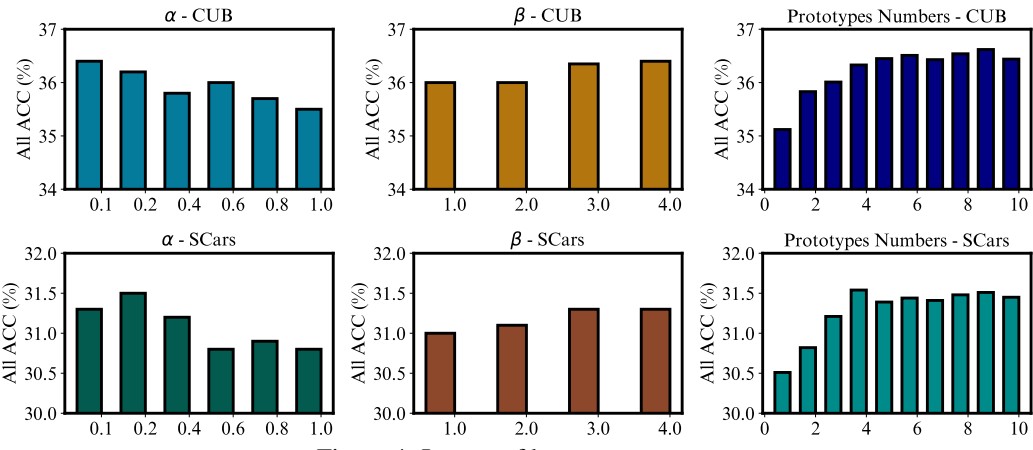

Figure 4: Impact of hyper-parameters.

Regarding $\beta$, the results are generally stable; however, higher values show improved performance across both datasets. Consequently, we selected $\alpha = 0.1$ and $\beta = 3$ for all datasets during training. 2) Additionally, we examine the impact of the number of prototypes per class on the CUB and Stanford Cars datasets, as shown in Fig. 4. When only one prototype per class is used, performance is suboptimal, as it fails to effectively represent the complexity of a category. For instance, a single bird species might exhibit different behaviors, such as flying or standing, that are not adequately captured by a single prototype. More prototypes allow for a better expression of the nuances within a category, which is crucial in fine-grained classification. Therefore, we opt to utilize 10 prototypes per class across all datasets.

**Hash Centers Analysis.** We conduct an analysis of the distribution of hash centers before and after training to further evaluate the impact of the proposed hash center optimization loss, $\mathcal{L}_c$. Specifically, we analyze the Hamming distances between hash centers on the CUB dataset. As depicted in Fig. 5, prior to training, the hash centers, derived from randomly initialized prototypes, are distributed relatively uniformly. Notably, some centers exhibit a Hamming distance of zero, indicating multiple centers sharing a single hash code. After training, the Hamming distance between all hash centers is at least $d_{max}$. This significant improvement demonstrates the effectiveness of $\mathcal{L}_c$ in ensuring that multiple categories do not share identical hash codes or reside excessively close to one another.

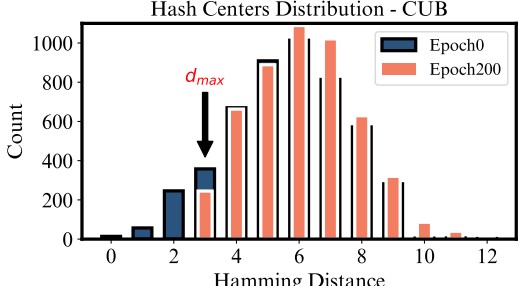

Figure 5: Evolution of hash centers distribution during the training process.

## 5 Conclusion

In this paper, we introduce a Prototypical Hash Encoding (PHE) framework for fine-grained On-the-fly Category Discovery. Addressing the limitations of existing methods, which struggle with the high sensitivity of hash-form category descriptors and suboptimal feature representation, our approach incorporates a prototype-based classification model. This model facilitates robust representation learning by developing multiple prototypes for each fine-grained category. We then map these category prototypes to corresponding hash centers, optimizing image hash features to align closely with these centers, thereby achieving intra-class compactness. Additionally, we enhance inter-class separation by maximizing the distance between hash centers, guided by the Gilbert-Varshamov bound. Experiments on eight fine-grained datasets demonstrate that our method outperforms previous methods by a large margin. Moreover, a visualization study is provided to understand the underlying mechanism of our method.

**Acknowledgement.** This work has been supported by the National Natural Science Foundation of China (62402157), the MUR PNRR project FAIR (PE00000013) funded by the NextGenerationEU, the EU Horizon project ELIAS (No. 101120237), the MUR PNRR project iNEST-Interconnected Nord-Est Innovation Ecosystem (ECS00000043) funded by the NextGenerationEU, and the EU Horizon project AI4Trust (No. 101070190).

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

# Appendix

## A   Implementation Details

### A.1   Datasets Details and Evaluation Metric Details

**Dataset Details.** As outlined in Table 6, our method is evaluated across multiple benchmarks. We have introduced the iNaturalist 2017 [35] dataset to the On-the-Fly Category Discovery (OCD) task, demonstrating our method's effectiveness in handling challenging fine-grained datasets. The iNaturalist 2017 dataset, sourced from the citizen science website iNaturalist, consists of 675,170 training and validation images across 5,089 natural fine-grained categories, which include Plantae (plants), Insecta (insects), Aves (birds), and Mammalia (mammals), among others, spread across 13 super-categories. These super-categories exhibit significant intra-category variations, making each a challenging dataset for fine-grained classification. For our evaluation, we selected Fungi, Arachnida, Animalia, and Mollusca as the four super-categories. Following the protocol established in OCD [12], the categories within each dataset are divided into subsets of seen and unseen categories. Specifically, 50% of the samples from the seen categories are used to form the labeled training set $\mathcal{D}_S$, while the remainder forms the unlabeled set $\mathcal{D}_Q$ for on-the-fly testing.

Table 6: Statistics of datasets used in our experiments.

|  | CUB | Scars | Pets | Food | Fungi | Arachnida | Animalia | Mollusca |
|---|---|---|---|---|---|---|---|---|
| $\|Y_S\|$ | 100 | 196 | 38 | 101 | 121 | 56 | 77 | 93 |
| $\|Y_Q\|$ | 200 | 98 | 19 | 51 | 61 | 28 | 39 | 47 |
| $\|\mathcal{D}_S\|$ | 1.5K | 2.0K | 0.9K | 19.1K | 1.8K | 1.7K | 1.5K | 2.4K |
| $\|\mathcal{D}_Q\|$ | 4.5K | 6.1K | 2.7K | 56.6K | 5.8K | 4.3K | 5.1K | 7.0K |

**Evaluation Metric Details.** We employ clustering accuracy, specifically the Strict-Hungarian method as outlined in OCD [12], as our evaluation protocol, a common choice in NCD/GCD tasks. Clusters are ranked by size, and only the top-$|Y_Q|$ clusters are retained; others are deemed misclassified. It

is crucial to outline our accuracy calculation for old and new classes. We perform the Hungarian assignment once for all categories $\mathcal{Y}_Q$, and subsequently measure classification accuracy for the "Old" and "New" subsets. This setup follows the protocols in OCD [12] and GCD [12].

## A.2 Training Details

**Model Training Details.** We employ the AdamW optimizer during training, using a learning rate of 1e-4 for the backbone and 1e-3 for the projection head and prototype layer, with a weight decay of 0.05. We train the model for 200 epochs. The batch size is uniformly set at 128 across all datasets for consistent comparisons with leading methods. Training incorporates the use of Exponential Moving Average (EMA). Experiments were conducted using Tesla V100 and 1080Ti GPUs, with results reported as the mean of three runs.

**Calculated $d_{\max}$.** The value of $d_{\max}$ is calculated using Eq. 7 for all datasets and hash code lengths $L$. The specific values of $d_{\max}$ are reported as follows:

Table 7: Values of $d_{\max}$ for different hash code lengths $L$ and datasets.

| $L$ | CUB | Scars | Pets | Food | Fungi | Arachnida | Animalia | Mollusca |
|-----|-----|-------|------|------|-------|-----------|----------|----------|
| 12bit | 3 | 3 | 4 | 3 | 3 | 4 | 3 | 3 |
| 16bit | 4 | 4 | 5 | 5 | 4 | 5 | 5 | 5 |
| 32bit | 10 | 10 | 12 | 11 | 11 | 11 | 11 | 11 |
| 64bit | 24 | 24 | 27 | 25 | 25 | 26 | 25 | 25 |

## A.3 Compared Methods Details

The setup for the comparative experiments follows OCD [12]. Detailed information is provided as follows: **Ranking Statistics (RankStat).** [38] AutoNovel utilizes Ranking Statistics to analyze sample relationships, particularly by using the top-3 indices of feature embeddings as category descriptors. This method aligns with the On-the-Fly Category Discovery (OCD) settings and poses a strong challenge to hash-based descriptors. For a fair comparison, we use the same backbone (DINO-ViT-B-16) for Ranking Statistics (RS) and retain only the fully-supervised learning stages, as no additional data can be used in the On-the-Fly Category Discovery (OCD) task. The embedding dimension is set to 32, resulting in a prediction space of $C_{32}^3 = 4,906$, comparable to our method and SMILE, which uses a hash code length of $L = 12$ and achieves a prediction space of $2^{12} = 4,096$. **Winner-take-all (WTA).** [39] To address potential biases towards salient features by Ranking Statistics, Winner-take-all (WTA) hash was proposed as an alternative. WTA avoids reliance on the global order of feature embeddings and instead utilizes indices of maximum values within divided feature groups. With a 48-dimension embedding divided into three groups, WTA generates a prediction space of $16^3 = 4096$, ensuring comparability for fair assessment. **Sequential Leader Clustering (SLC).** [37] We employ the same backbone and conducts fully supervised training on the support set for SLC. For on-the-fly testing, SLC utilizes features extracted from the backbone on the query set. We optimized the hyperparameters based on the CUB dataset, applying these uniformly across other datasets to maintain consistency in our comparisons.

## A.4 Pseudo-code

We have detailed the description of our PHE framework and the Hamming Ball Based Model Inference in the main text. The pseudo-code for the Hamming Ball Based Model Inference is provided in Algorithm 1, and the training process of our PHE framework is shown in Algorithm 2.

**Algorithm 2:** Pseudocode for PHE

**Input:** Training data (support set) $\mathcal{D}_S$ contains labeled data, image encoder $\mathcal{E}(\cdot)$, a linear projection head $\mathcal{H}_f(\cdot)$, a prototype layer $g_\mathbf{p}(\cdot)$, a frozen layer $FC$ and a linear projection head $\mathcal{H}_h(\cdot)$, hyper parameters {training epochs $E_1$, prototypes per class $k$, et al.}.

1 /* Model Training */ ;
2 **for** *each epoch* $e = 1 \ldots E_1$ **do**
3    **for** *each batch* $(\mathbf{x}_i, \boldsymbol{y}_i) \in \mathcal{B}$ **do**
4       /* CPG module */ ;
5       $\mathbf{z}_i = \mathcal{H}_f(\mathcal{E}(\mathbf{x}_i))$ ;                                 // image feature
6       $\mathbf{s}_i = g_\mathbf{p}(\mathbf{z}_i)$ ;                              // similarity score
7       Compute $\mathcal{L}_p$ by Eq. (2) on $\mathbf{s}_i$ and $FC$;
8       Trained prototypes $P_{c_j}$ for class $c_j$ ;              // prototypes
9       /* DHC module */ ;
10      $\mathbf{h}_j = \mathcal{H}_h(\sum P_{c_j}/k)$ ;                  // hash centers
11      $\mathbf{b}_i = \mathcal{H}_h(\mathbf{z}_i)$ ;                      // image hash feature
12      Compute $\mathcal{L}_f$ by Eq. (3) on $\mathbf{b}_i$ and all hash centers $\mathbf{h}$;
13      Compute $d_{max}$ by Eq. (7) ;      // hash centers optimization upper bond
14      Compute $\mathcal{L}_{sep}$ by Eq. (5) on $d_{max}$ and all hash centers $\mathbf{h}$;
15      Compute $\mathcal{L}_q$ by Eq. (8) on all hash centers $\mathbf{h}$;
16      Compute $\mathcal{L}_c$ by Eq. (9) on $\mathcal{L}_{sep}$ and $\mathcal{L}_q$ ;    // hash center optimization loss
17      Compute the overall loss $\mathcal{L}$ by Eq. (10);
18      Back-propagation and optimize $\mathcal{E}, \mathcal{H}_f, g_\mathbf{p}, \mathcal{H}_h$;

19 **return** Trained model PHE$(\cdot)$ ;

## B   Additional Experiment Results and Analysis

### B.1   Error Bars for Main Results

In the main paper, we present the full results as the average of three runs to mitigate the impact of randomness. Detailed outcomes for our PHE, encompassing mean values and population standard deviation, are delineated in Table 8.

Table 8: Mean and std of accuracy in three independent runs.

| Dataset | All | Old | New |
|---|---|---|---|
| CUB [31] | 36.4±0.17 | 55.8±1.58 | 27.0±0.98 |
| Stanford Cars [32] | 31.3±0.49 | 61.9±1.29 | 16.8±0.41 |
| Oxford Pets [33] | 48.3±0.96 | 53.8±3.24 | 45.4±1.65 |
| Food101 [34] | 29.1±0.25 | 64.7±0.45 | 11.1±0.62 |
| Fungi [35] | 31.4±0.39 | 67.9±1.76 | 15.2±0.45 |
| Arachnida [35] | 37.0±0.34 | 75.7±0.33 | 12.6±0.52 |
| Animalia [35] | 40.3±0.40 | 55.7±2.16 | 31.8±1.25 |
| Mollusca [35] | 39.9±0.66 | 65.0±2.42 | 26.5±1.27 |

### B.2   Inference on Different Input Sequences

We evaluated the accuracy of our PHE method on CUB and Stanford Cars under different input sequences, as shown in Table 9. In the table, "fixed Sequences" indicates that the test data order is not shuffled (all results and comparisons in the main paper are tested this way), while "Random Sequences" indicates results obtained by randomly shuffling the test data order, with the results averaged over 10 runs. As can be seen from the table, our results are very stable and not affected by the input sequence order. This stability is mainly due to our hash center optimization, which ensures significant inter-class separation, and the use of Hamming balls based on hash centers for category prediction, which is very robust to different input sequences.

Table 9: Results with inference on different input sequences. The best results are marked in **bold**.

| Methods | CUB | | | SCars | | |
|---|---|---|---|---|---|---|
| | All | Old | New | All | Old | New |
| Fixed Sequences | 36.4 | 55.8 | **27.0** | 31.3 | 61.9 | **16.8** |
| Random Sequences | **36.5** | **55.9** | **27.0** | **31.4** | **62.1** | 16.7 |

## B.3 Impact of Feature Dimension on SMILE Performance

We analyze the performance of the SMILE method [12] with different feature dimensions in Table 10. In the table, "Offline" and "Clustering Acc" refer to the clustering accuracy obtained by applying k-means clustering on the features of all test data, given a fixed number of categories (200 for CUB). From the table, it is evident that as the feature dimension $L$ decreases, the on-the-fly prediction accuracy significantly improves. This is because the "high sensitivity" issue of hash-form category descriptors becomes more pronounced at higher feature dimensions. On the other hand, as the feature dimension decreases, there is a notable reduction in offline clustering accuracy, with a decrease of 9.9% when reducing $L$ from 256 to 12. This indicates that when the feature dimension is too low, it inevitably damages the model's ability to distinguish categories, resulting in poorer feature representation learning. Therefore, our PHE framework separates feature learning from category representation learning, which is one of the reasons why our PHE framework performs better.

Table 10: Online vs. offline predictions with different hash code lengths $L$ in SMILE on CUB.

| $L$ | On-the-fly | | | Offline |
|---|---|---|---|---|
| | All | Old | New | Clustering Acc |
| 12bit | **32.2** | **50.9** | **22.9** | 38.7 |
| 64bit | 22.6 | 45.3 | 11.2 | 45.2 |
| 128bit | 17.3 | 38.2 | 6.8 | 47.4 |
| 256bit | 13.2 | 28.4 | 5.4 | **48.6** |

## B.4 Comparison with Deep Hash Methods

We conduct comparative experiments using various deep hashing methods based on our prototype learning model across the CUB and Stanford Cars datasets. The results, as detailed in Table 11, demonstrate the effectiveness of our PHE. Traditional methods like DPN, OrthoHash, and CSQ generate suboptimal hash centers, primarily designed for retrieval tasks with a focus on instance-level discrimination. Unlike these methods, our PHE incorporates the Gilbert-Varshamov bound to guide the learning of hash centers, specifically for category discovery. This strategy ensures that the hash centers are not only discriminable but also preserve the rich category-specific information inherent in category-level prototypes. Furthermore, our PHE outperforms MDSH, which optimizes hash codes to align with pre-defined static hash centers. In contrast, our approach dynamically maps hash centers from each category's prototypes, continuously refining these through end-to-end optimization. Additionally, our design incorporates a Hamming-ball-based inference mechanism, which significantly reduces hash sensitivity. For instance, when the hash code length $L$ is set to 32, the MDSH-Hamming ball configuration outperforms the standard MDSH by an average margin of 5.9% across the two datasets.

## B.5 Computational Cost Analysis

The computational cost of PHE is relatively low and primarily depends on the number of known categories, rather than directly correlating with the scale of the dataset. Specifically, each known category is associated with a hash center, with hash centers $\mathbf{h}$ of shape [$class\ nums, code\ length$]. The main computational operations involve straightforward two-dimensional matrix multiplications. These include the dot product of features $\mathbf{b}$, with a shape [$batch\ size, code\ length$], and hash centers, as well as the dot product operations used in calculating Hamming distances between hash centers.

Table 11: Comparative results of various hashing methods on CUB and Stanford Cars datasets using 12-bit and 32-bit hash code lengths. "MC" denotes manually obtained centers compliant with the Gilbert-Varshamov bound.

| Methods | CUB 12bit | | | CUB 32bit | | | SCars 12bit | | | SCars 32bit | | |
|---|---|---|---|---|---|---|---|---|---|---|---|---|
| | All | Old | New | All | Old | New | All | Old | New | All | Old | New |
| DPN [20] | 22.2 | 38.0 | 14.2 | 12.5 | 11.1 | 13.2 | 18.8 | 36.1 | 10.5 | 12.8 | 20.4 | 9.1 |
| OrthoHash [21] | 30.0 | 49.2 | 20.5 | 13.6 | 24.8 | 8.0 | 19.6 | 37.2 | 11.1 | 13.2 | 20.0 | 9.8 |
| CSQ [22] | - | - | - | 26.1 | 45.3 | 16.5 | - | - | - | 23.1 | 44.1 | 13.0 |
| CSQ-MC | - | - | - | 26.4 | 50.6 | 14.3 | - | - | - | 26.9 | 61.8 | 10.0 |
| MDSH [23] | 34.3 | **57.6** | 22.8 | 27.4 | 40.8 | 20.7 | 28.8 | 60.2 | 13.7 | 25.8 | 47.8 | 15.2 |
| MDSH+Hamming Ball | 35.1 | 55.0 | 25.3 | 35.5 | 47.8 | **29.3** | 29.8 | 56.1 | **17.1** | 29.5 | 56.2 | **16.6** |
| Ours | **36.4** | 55.8 | **27.0** | **38.5** | **59.9** | 27.8 | **31.3** | **61.9** | 16.8 | **31.5** | **64.0** | 15.8 |

As detailed in Table 12, although the Food dataset is larger than both CUB and Stanford Cars, it includes only 100 categories. Consequently, the average training time per sample for the Food dataset is significantly lower than that observed for the CUB and Stanford Cars datasets.

Table 12: Comparison of training time per sample across CUB, Stanford Cars and Food datasets.

| Dataset | CUB#200 | SCars#198 | Food#101 |
|---|---|---|---|
| Number of training samples | 1.5k | 2.0k | 19.1k |
| Training time / minute | 100.22 | 161.37 | **691.39** |
| Training time per sample / second | 4.01 | 4.84 | **2.17** |

## B.6 Training Efficiency Analysis

We provide a comparison of training times between our PHE and the state-of-the-art method, SMILE, with results shown in Table 13. To ensure fairness, all experiments are conducted on an NVIDIA RTX A6000 GPU, with both algorithms trained over 200 epochs using mixed precision. The dataloader parameters are kept consistent across tests, with a batch size of 128. According to the table, our average training time across four datasets is 45.8 minutes shorter than that of SMILE. This improvement is primarily due to the higher computational demands of SMILE's supervised contrastive learning approach, which processes two different views of samples for representation learning.

Table 13: Comparison of training times (in minutes)

| Method | CUB | SCars | Food | Pets |
|---|---|---|---|---|
| SMILE | 127.70 | 177.54 | 819.93 | 80.37 |
| PHE (ours) | **100.22** | **161.37** | **691.39** | **69.48** |

## B.7 Results of Different Dataset Splits

We conduct experiments using different proportions of old category selection on the CUB and Stanford Cars datasets, as shown in Table 14, with all accuracy results reported and hash code length $L$ set to 12. Our PHE outperforms SMILE by an average of 5.95% when 75% of old categories are selected and by 1.0% when 25% are selected. This indicates that our PHE better models the nuanced inter-category relationships in fine-grained category divisions as the number of categories increases.

## B.8 Compared to Prototype Learning Method

We conduct comparative experiments using the powerful prototype learning method SimGCD [13], based on our category encoding method. Due to the lack of unlabeled data, we remove the unsupervised loss component, $L_{cls}^u$, and the prototypes corresponding to new categories in SimGCD. As shown in Table 15, our use of SimGCD for prototype learning, mapping the prototypes learned by

Table 14: Comparison with different known category split percentages.

| Method | CUB-25% | CUB-50% | CUB-75% | SCars-25% | SCars-50% | SCars-75% |
|---|---|---|---|---|---|---|
| SMILE | 19.9 | 32.2 | 41.2 | 12.6 | 26.2 | 37.0 |
| PHE (ours) | **21.2** | **36.4** | **46.5** | **13.3** | **31.3** | **43.6** |

SimGCD to hash centers for category encoding, yields very poor results on both datasets. "SimGCD-MC", which employs manually obtained centers satisfying the Gilbert-Varshamov bound along with features from the SimGCD projection head, shows an average improvement of 8.8% across the two datasets. This performance boost demonstrates that the prototypes learned by SimGCD are not suitable for category encoding in fine-grained scenarios. Our PHE offers two main advantages. First, unlike SimGCD, which learns only one prototype per category, our PHE generates multiple prototypes per class, effectively modeling intra-class variance of fine-grained categories. Second, unlike the prototypes in SimGCD, which serve as classifier weights, the prototypes in PHE can be explicitly visualized, providing additional insights into the model's behavior.

Table 15: Comparative results of different representation learning methods on CUB and Scars datasets. "MC" denotes manually obtained centers compliant with the Gilbert-Varshamov bound.

| Method | CUB | | | SCars | | |
|---|---|---|---|---|---|---|
| | All | Old | New | All | Old | New |
| SimGCD [13] | 25.0 | 49.9 | 12.6 | 21.9 | 38.5 | 13.9 |
| SimGCD-MC | 34.1 | **60.6** | 20.8 | 30.3 | **65.9** | 13.0 |
| PHE (ours) | **36.4** | 55.8 | **27.0** | **31.3** | 61.9 | **16.8** |

## C  Broader Impact and Limitations Discussion

**Broader Impact.** The On-the-fly Category Discovery (OCD) task is designed to learn category differences from observed categories and then make real-time predictions across a broader range, including unknown categories. Our proposed Prototypical Hash Encoding (PHE) method for the OCD task holds potential for application in open-world scenarios. For instance, it can assist in botanical classification, where new plant species are discovered and need to be quickly integrated into existing categories without retraining the entire system.

**Limitations.** Despite the superior performance of our Prototypical Hash Encoding (PHE) framework compared to existing methods, it still requires further research to enhance the accuracy of recognizing unknown categories. Current On-the-fly Category Discovery (OCD) methods generally achieve low accuracy in predicting new categories, as these categories were not encountered during training, posing significant challenges to effective generalization. Future research should focus on improving the model's ability to recognize and categorize new, unseen categories.

**A Possible Solution in Future Work.** Due to the constraints of training data in the OCD setting, we are considering the integration of additional knowledge from pre-trained Large Language Models (LLMs). First, we can utilize LLMs to establish a bank of category attribute prototypes, which are expected to be relevant across both known and unknown categories. Subsequently, during the real-time prediction process, we plan to employ LLMs and Vision-Language Pretrained Models (VLMs) to match these attribute prototypes with unknown categories. Ultimately, by integrating both instance and attribute features, we aim for our PHE to generate more precise predictions.

## D  Additional Visualization Analysis

We have conducted additional visualization analysis with images from the CUB dataset and Stanford Cars dataset, as shown in Fig. 6 to 11. As discussed in Sec. 4.4, in addition to using hash codes to represent categroies, our PHE framework allows for visual analysis from the perspective of prototype similarity to understand why certain images are identified as new classes, as depicted in Fig. 6, 7, 9, and 10. Furthermore, even when images are misclassified, such as assigning an image from a new category to an old category as shown in Fig. 8 and Fig. 11, we can still gain interesting insights from the prototype perspective. Firstly, these images from new categories indeed exhibit high similarity

to the category where the activated prototypes belong to. Secondly, although the images from new categories show high similarity to old ones, the computed similarity is relatively lower compared to samples that truly belong to that old category. For example, in Fig. 8, the similarity values are 5.48 versus 9.14 for the misclassified new category and the true sample of the old category, respectively.

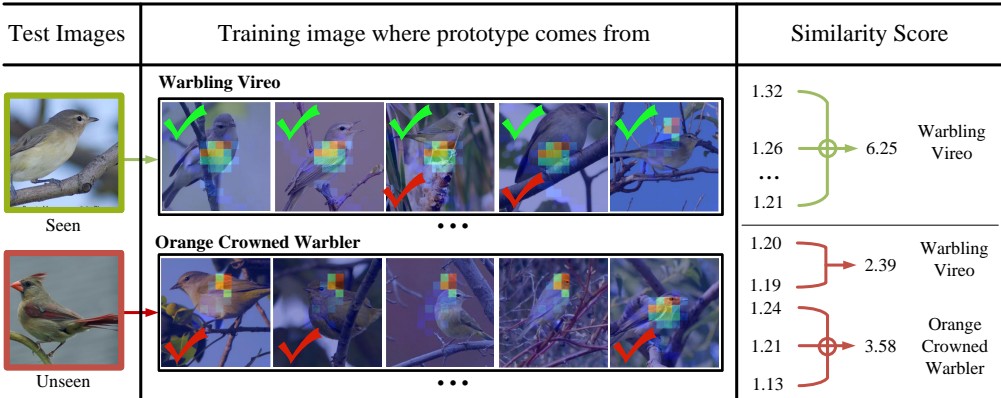

Figure 6: Case Study of the Cardinal and the Warbling Vireo.

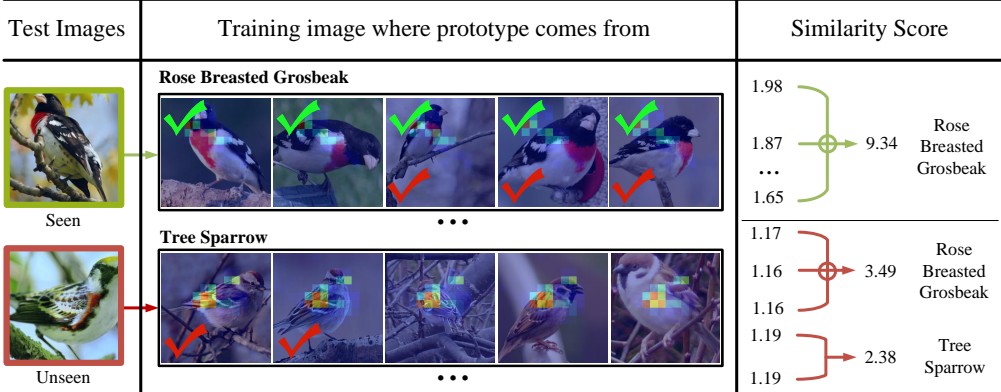

Figure 7: Case Study of the Chestnut sided Warbler and the Rose breasted Grosbeak.

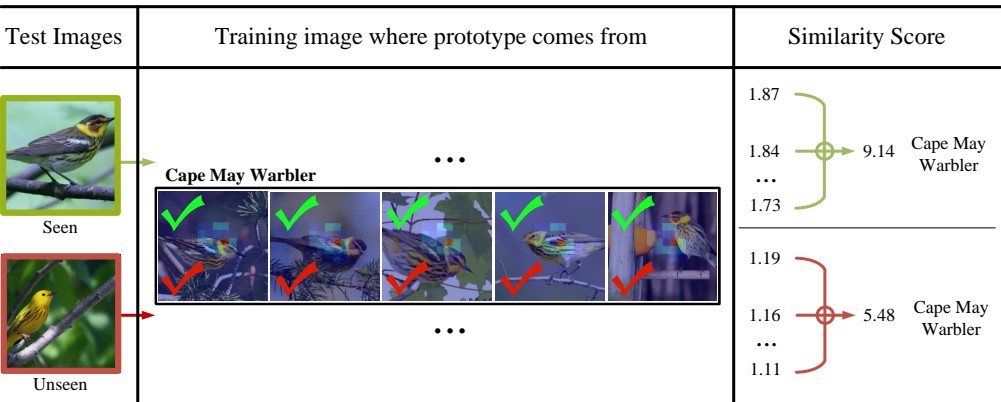

Figure 8: Case Study of the Yellow Warbler and the Cape May Warbler.

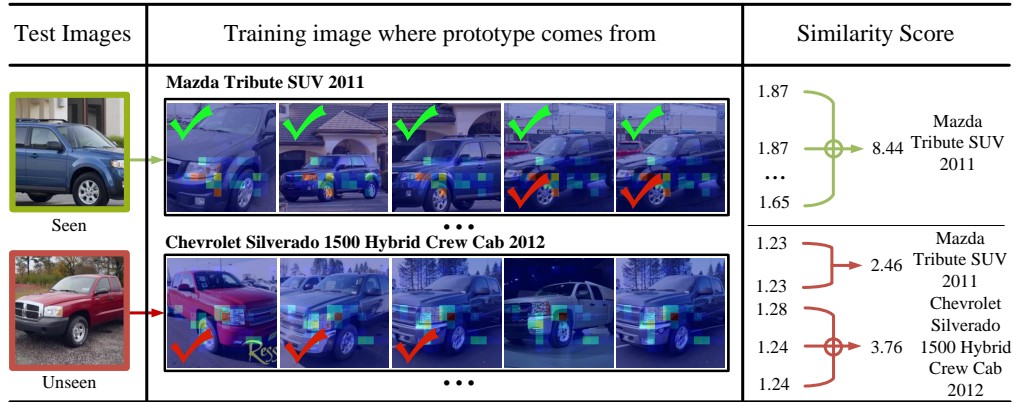

Figure 9: Case Study of the Dodge Dakota Club Cab 2007 and the Mazda Tribute SUV 2011.

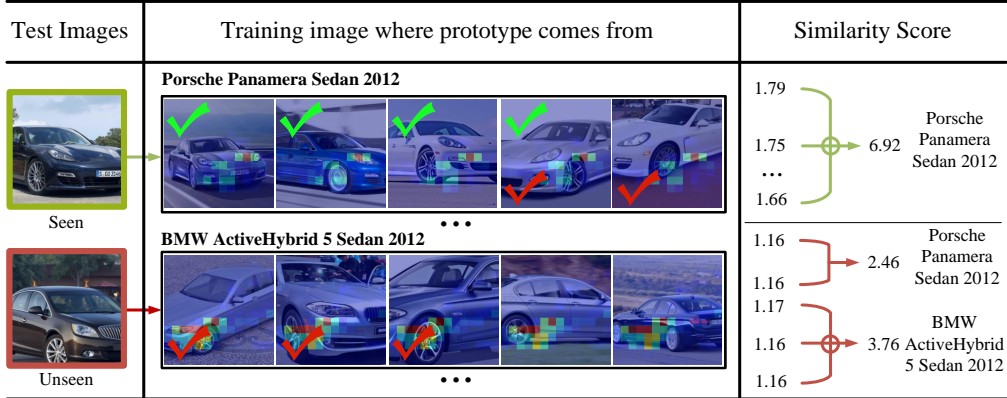

Figure 10: Case Study of the Buick Verano Sedan 2012 and the Porsche Panamera Sedan 2012.

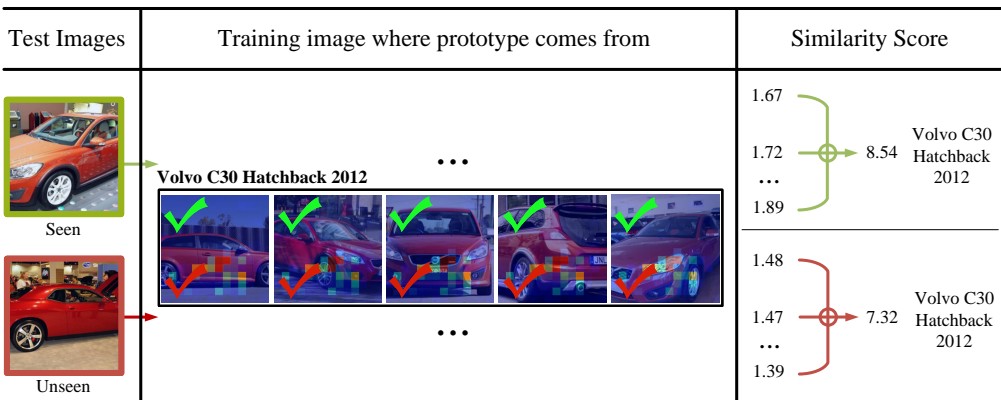

Figure 11: Case Study of the Dodge Challenger SRT8 2011 and the Volvo C30 Hatchback 2012.

