# OpenReview forum: "Prototypical Hash Encoding for On-the-Fly Fine-Grained Category Discovery"
_NeurIPS.cc/2024/Conference — NeurIPS 2024 poster_

### Official Review · Reviewer_dwc8 · 2024-06-24

**Soundness:** 3
**Presentation:** 3
**Contribution:** 2
**Rating:** 5
**Confidence:** 4

**Summary:**

This paper proposes a prototypical deep hashing framework to address the fine-grained on-the-fly category discovery problem. The proposed method includes two main loss functions: first, distance minimization between the encoded hash features to the category-representative hash coding after projection $\mathcal{H}_h$; second, enforcing minimal distance separation among hash encoding of different categories, which are enforced to be quantized to -1 and +1 for each dimension. The experimental results demonstrate the effectiveness of each loss function. The performance gains are considerable on all benchmarks compared with previous state-of-the-art baselines.

**Strengths:**

- The performance gains of the proposed approach are considerable.
- The proposed method is robust to the varying coding length.
- The identified ‘sensitivity issue’ problem exists for on-the-fly category discovery.

**Weaknesses:**

- Writing: It is informal to leave all the related works in the appendix, which will confuse the readers on the contribution of this work. Besides, some expressions are not
- Novelty: The technical contribution is limited in that the prototypical learning [3] is a mature practice in category discovery and the author seems to adapt the deep hashing method in [4] to this problem.
- Motivation: The motivation of achieving the balance between instance discrimination and class discrimination, especially with prototypical learning in the category discovery field is not new [2,3]. However, this is accepted to some extent since on-the-fly category discovery is a new problem.

**Questions:**

- It would be clearer if the author could mention the technical practice of prototype-based visualization examples, though it may be proposed by the previous work, ProtopFormer.
- In Table 3, how will the semi-supervised contrastive loss perform [1] in addition to the supcon and unsupcon baselines? These results matter to solidify the motivation for using the deep hashing framework.


[1] Generalized Category Discovery

[2] PromptCAL: Contrastive Affinity Learning via Auxiliary Prompts for Generalized Novel Category Discovery

[3] Parametric Classification for Generalized Category Discovery: A Baseline Study

[4] Deep hashing with minimal-distance-separated hash centers

**Limitations:**

As mentioned by the author, on-the-fly category discovery is a novel and challenging problem, therefore existing methods do not achieve satisfying performance on real-world tasks.

---

> ### Author Rebuttal · Authors · 2024-08-07
>
> We sincerely thank the reviewer’s recognition of our method’s good performance and its robustness to varying coding lengths (sensitivity issue). We value the reviewer’s insightful comments and will incorporate these into our final revision.
>
> **Q1: Writing**. Thanks for your constructive suggestions. We will **leave the section of related works to the main text**, and carefully check and improve the expressions in the revision.
>
> **Q2: Technical contribution**.
>
> We argue that our contribution mainly lies in identifying and effectively mitigating the high sensitivity issue existing in current hash-based OCD methods, which was acknowledged by Reviewers 6xA4 and Pa9u as well. We provide elaborated discussions as follows.
>
> **1. Compared to prototype learning method (SimGCD [1])**, our PHE has two main advantages. **First**, unlike SimGCD that learns only one prototype for one category, our PHE generates multiple prototypes for one class, which is favorable for modeling intra-class variance of fine-grained categories. Empirically, we have supplemented experiments with the prototype learning method used in SimGCD to validate the superiority of our methods as shown in the table below.
>
> | Dataset       |          | CUB      |          |          | SCars    |          |
> | ------------- | -------- | -------- | -------- | -------- | -------- | -------- |
> | Method        | All      | Old      | New      | All      | Old      | New      |
> | SimGCD+PHE    | 25.0     | 49.9     | 12.6     | 21.9     | 38.5     | 13.9     |
> | SimGCD-MC+PHE | 34.1     | **60.6** | 20.8     | 30.3     | **65.9** | 13.0     |
> | PHE (ours)     | **36.4** | 55.8     | **27.0** | **31.3** | 61.9     | **16.8** |
>
> Specifically, due to lack of unlabeled data, we removed the $L_{cls}^u$ and the prototypes corresponding to new categories in SimGCD. In the table, "SimGCD+PHE" indicates that we used SimGCD for prototype learning while also mapping the prototypes learned by SimGCD to hash centers for category encoding. This approach yielded very poor results on both datasets. "SimGCD-MC+PHE" refers to the use of manually obtained centers that satisfy the Gilbert-Varshamov bound and features from the SimGCD projection head for category encoding. Compared to mapping SimGCD’s prototypes to hash centers, the SimGCD-MC+PHE variant shows an average improvement of 8.8% on two datasets, demonstrating that the prototypes learned by SimGCD are not suitable for category encoding in fine-grained OCD scenarios.
> **Second**, Unlike the prototypes in SimGCD which are weights of classifiers, the learned prototype in PHE can be explicitly visualized, which provides additional perspective for interpreting the model's behavior, as illustrated in Fig.3 of the main text.
>
> **2. Compared with deep hash method (MDSH[2]),** our PHE has two main differences: **First**, unlike MDSH that pre-calculates hash centers and fixes these centers during training, our hash centers are derived by mapping from category prototypes and then are updated by end-to-end optimization, which better preserves the relationships between fine-grained categories learned in the prototype feature space. **Second**, we design an additional Hamming ball-based inference tailor-made for OCD, which effectively mitigates the sensitivity issues associated with using hash codes. Furthermore, we have supplemented experiments and the results in the following table verify the superiority of our PHE. We also provide the results of more hash-based baselines in **Q1** of Reviewer **6xA4.**
>
> | Datasets  |          | CUB      |          |          | SCars    |          |
> | --------- | -------- | -------- | -------- | -------- | -------- | -------- |
> | Methods   | All      | Old      | New      | All      | Old      | New      |
> | MDSH[2]   | 34.3     | **57.6** | 22.8     | 28.8     | 60.2     | 13.7     |
> | PHE (ours) | **36.4** | 55.8     | **27.0** | **31.3** | **61.9** | **16.8** |
>
> **Q3: Motivation of instance discrimination and class discrimination**
>
> We agree that both OCD and GCD [3] need to learn good representations with “optimal balance between instance discrimination and class discrimination”. However, our primary motivation comes from the identification of the high sensitivity issue caused by applying hash codes in OCD tasks, which significantly hinders the effectiveness of the current hash-based OCD method. Motivated by this, we present the specific prototype design, loss implementation and Hamming-ball-based inference, to mitigate the high-sensitivity issue.
>
> **Q4: Technical practice of prototype-based visualization examples**. The process mainly consists of three steps: 1) Input an image to obtain its feature representation, while also capturing the attention map during forward propagation. 2) Select the top-k most similar (activated) prototypes from all prototypes. 3) Visualize the original samples/images corresponding to these prototypes. We will include detailed examples and process descriptions in the revision.
>
> **Q5: Will the semi-supervised contrastive loss (SSCL) perform?** Firstly, we'd like to clarify that in the OCD setting, only labeled data of known classes is available for model training, and unlabeled data appears only in an on-the-fly format during testing. Thus, the SSCL used in GCD[1] is incompatible with OCD. As for the experiments in Tab. 3, in fact, we did not include experiments with "supcon and unsupcon baselines". “Supcon Cls” represents the use of classification methods based on supervised contrastive learning for representation learning. According to the results, we find that although this variant performs well on seen categories, their generalization capabilities are inferior to our full prototype-based method.
>
> [1] Parametric Classification for Generalized Category Discovery: A Baseline Study. ICCV 2023
>
> [2] Deep Hashing with Minimal-Distance-Separated Hash Centers. CVPR 2023.
>
> [3] Generalized Category Discovery. CVPR 2022.

---

> > ### Comment · Reviewer_dwc8 · 2024-08-09
> >
> > Thanks for the reply. Most of my questions are well addressed. Although the author clarified their contributions and differences compared with previous works, my concern about the technical novelty and contribution of this work has been partly addressed. After rebalancing its weaknesses and strengths, I decide to raise my score by one.

---

> > > ### Author Response · Authors · 2024-08-09
> > >
> > > Dear Reviewer dwc8,
> > >
> > > We greatly appreciate your satisfaction with our responses, and very glad you increase the rating! We will add the above important discussions in the final manuscript and highlight them.
> > >
> > > Thanks again for your valuable suggestions and comments. We enjoy communicating with you and appreciate your efforts!

---

### Official Review · Reviewer_ZQiT · 2024-07-02

**Soundness:** 3
**Presentation:** 3
**Contribution:** 2
**Rating:** 5
**Confidence:** 4

**Summary:**

This paper introduces a novel framework called Prototypical Hash Encoding (PHE) for On-the-fly Category Discovery (OCD), which aims to discover both known and unknown categories from streaming data using labeled data of known categories. PHE first learns many prototypes for each category and then maps the learned prototypes to hash codes to distinguish samples from known or novel categories with a threshold.

**Strengths:**

1. The paper is well-written and easy to follow.
2. The proposed method addresses the limitations of previous hash-based OCD models by reducing sensitivity and preserving discriminative information through prototypes.
3. The proposed method achieves significant improvements in accuracy across various datasets.

**Weaknesses:**

1. The proposed method is an improvement from the previous work [1], sharing the same core idea of utilizing hash codes for OCD, which limits the novelty of the paper.
2. Despite the improved performance, the reasons why prototypes can address the sensitivity issue are not analyzed in depth.
3. Compared with feature-level prototypes, the advantages of hash code-based category prototypes have not been demonstrated or experimentally verified.

[1] Ruoyi Du, Dongliang Chang, Kongming Liang, Timothy Hospedales, Yi-Zhe Song, and Zhanyu Ma. On-the-fly category discovery. In Proceedings of the IEEE/CVF Conference on Computer Vision and Pattern Recognition, pages 11691–11700, 2023.

**Questions:**

1. Compared with feature-level prototypes or prototypes after dimensionality reduction, what are the advantages of hash code-based prototypes?
2. How is the training efficiency of the model compared to SMILE?
3. How are hyperparameters selected without the validation set?

---

> ### Author Rebuttal · Authors · 2024-08-07
>
> We sincerely thank the reviewer’s recognition of our motivation to address the limitations of previous OCD models. We value the reviewer’s insightful comments and will incorporate these into our final revision.
>
> **Q1: Sharing the same core idea of utilizing hash codes for OCD**. We indeed use hash codes as category descriptors, which are also widely used in image retrieval and deep hashing methods. However, we argue that our innovation lies in: 1. Identifying the high sensitivity issue when applying hash codes to the challenging fine-grained OCD task, where only data from known categories is available. 2. We have introduced a new OCD framework, PHE, that explicitly achieves inter-class separability and intra-class compactness, compared to the SOTA method SMILE. 3. Unlike SMILE, we designed an on-the-fly inference method based on the Hamming ball. Our PHE method effectively mitigates the high sensitivity problem, for example, surpassing SMILE by 15.5% in terms of all accuracy on the CUB dataset with hash code bits = 64. This point has also been acknowledged by Reviewers 6xA4 and Pa9u.
>
> **Q2: The reasons why prototypes can address the sensitivity issue**.
> Thanks for your insightful comments. We'd like to explain as follows.
> 1. In fact, instead of depending on only prototypes themselves, we mitigate the sensitivity issue by 1) constraining the hash centers (mapped from feature-level prototypes) to be at least a Hamming distance of $d_{max}$ apart based on the Gilbert-Varshamov bound; 2) representing a category using a Hamming ball of radius $max(\lfloor \frac{d_{max}}{2} \rfloor, 1)$.
>
> 2. The role of prototypes is to: 1) achieve better representation learning; 2) map the category prototypes to category hash centers, unifying representation learning with hash encoding. The importance of prototype learning is evident from the variant without prototype learning loss $L_f$ in Tab.2 of the main paper. Removing the prototype learning loss $L_f$ results in an obvious accuracy drop on datasets in terms of all, old and new accuracy.
>
> We will clarify this point more clearly in the introduction.
>
> **Q3: Feature-level prototypes vs.  Hash code-based prototypes**
> 1. **Experimental results with a powerful feature-level prototype learning method.**
> Firstly, we would like to further explain that in the OCD setting, test data appear one-by-one, and the OCD task requires the model to instantly assign a category to the test data. Therefore, if only feature-level prototype learning is used, the method needs an additional online clustering approach to yield real-time category descriptors, which is also acknowledged in the original OCD paper [1]. Thus, we adapt SimGCD [2] ( an advanced prototype learning approach for GCD, which was also acknowledged by Reviewer dwc8) into OCD settings and implement two variant approaches: 1) SimGCD attached a online clustering approach SLC in [1] to achieve instance-wise inference. 2) SimGCD with our hash prototype framework. The experimental results below show that introducing PHE significantly improved all accuracy across two datasets. Meanwhile, our full method achieves better generalization ability, especially on new categories.
>
> | Dataset                                         |          | CUB      |          |          | Scars    |          |
> | ----------------------------------------------- | -------- | -------- | -------- | -------- | -------- | -------- |
> | Method                                          | All      | Old      | New      | All      | Old      | New      |
> | SimGCD [2] (only feature-level prototype learning) | 27.1     | 50.7     | 15.3     | 21.5     | 39.9     | 12.5     |
> | SimGCD [2] + hash code-based prototype learning             | 34.1     | **60.6** | 20.8     | 30.3     | **65.9** | 13.0     |
> | PHE (ours)                        | **36.4** | 55.8     | **27.0** | **31.3** | 61.9     | **16.8** |
>
>
> 2. **Advantages of hash code-based features.** Hash code-based features can directly use the features’ signs (hash codes) as category descriptors, where features with the same category descriptor are considered to be of the same category. Therefore, compared to feature-level prototype learning, introducing a hash code-based design can directly optimize category descriptors, achieving higher accuracy in the OCD setting. According to the results in Table 1 of the main paper, where only feature-level prototype learning results (SLC) and hash code-based prototypes learning (our PHE) are compared, introducing hash code-based prototypes improved the average accuracy by 8.2% across eight datasets.
>
> **Q4: Training efficiency**. We provided a comparison of training times between our PHE and the SOTA method, SMILE, measured in minutes, as shown in the table below. To ensure fairness, all experiments were conducted on an NVIDIA RTX A6000 GPU. Both algorithms were trained for 200 epochs using mixed precision training. The dataloader parameters were consistent, with a batch size of 128 and num_workers set to 8. According to the table, our average training time across four datasets is less by 45.8 minutes compared to SMILE. This is primarily due to SMILE’s use of supervised contrastive learning with two views of samples for representation learning, which requires higher computational resources.
>
> | Method    | CUB    | Scars  | Food   | Pets  |
> | --------- | ------ | ------ | ------ | ----- |
> | SMILE     | 127.70 | 177.54 | 819.93 | 80.37 |
> | PHE (ours) | 100.22 | 161.37 | 691.39 | 69.48 |
>
> **Q5: Hyperparameter section.** To avoid over-hyperparameter tuning, we use a fixed set of hyperparameters which are obtained based on CUB to report accuracy for all datasets, without tuning them individually for each dataset.
>
> If we have misunderstood any of your concerns, please let us know in the future comments.
>
> [1] On-the-fly category discovery. CVPR 2023.
>
> [2] Parametric Classification for Generalized Category Discovery: A Baseline Study. ICCV 2023

---

> > ### Comment · Reviewer_ZQiT · 2024-08-10
> >
> > Thanks for your reply, my concerns and questions have been partly addressed. I will raise my score by one.

---

> > > ### Author Response · Authors · 2024-08-11
> > >
> > > Dear Reviewer ZQiT,
> > >
> > > Thank you for your positive feedback and for increasing the rating! We will include the important discussions you mentioned in the final manuscript.
> > >
> > > We truly enjoy our interactions and greatly appreciate all your efforts!

---

### Official Review · Reviewer_Pa9u · 2024-07-04

**Soundness:** 3
**Presentation:** 3
**Contribution:** 2
**Rating:** 6
**Confidence:** 3

**Summary:**

This paper addresses the On-the-fly Category Discovery (OCD) task, which involves utilizing existing category knowledge to recognize both known and unknown categories in new data streams in real-time. To tackle the high sensitivity and suboptimal feature representation issues of existing methods when dealing with fine-grained categories, the paper proposes an innovative Prototypical Hash Encoding (PHE) framework. This framework uses a Category-aware Prototype Generation (CPG) module to represent each fine-grained category with multiple prototypes and employs a probabilistic masking strategy to encourage the model to fully capture intra-class diversity. The Discriminative Category Encoding (DCE) module maps the generated category prototypes to low-dimensional hash centers, optimizing image hash features to ensure intra-class compactness and inter-class separation. Additionally, a center separation loss function is designed to maintain a minimum Hamming distance between different category hash centers. Experimental results on multiple datasets confirm the superiority of this method.

**Strengths:**

The proposed PHE method demonstrates its superiority over existing state-of-the-art methods across multiple datasets. It effectively addresses the issues of large intra-class variance and small inter-class variance, while minimizing the information loss associated with dimensionality reduction. The paper employs visualization techniques to analyze the underlying mechanism by which PHE groups samples into known or unknown categories. Moreover, due to its optimization based on hash centers and inference process based on Hamming balls, the PHE method shows stable performance across different hash code lengths, effectively mitigating the "high sensitivity" problem. In contrast to the SMILE method, which exhibits significant accuracy degradation and instability with increasing hash code lengths, the PHE method maintains remarkable stability and consistency.

**Weaknesses:**

Although the proposed Prototype Hashing Encoding (PHE) framework outperforms existing methods in terms of performance, further research is needed to improve its accuracy in recognizing unknown categories. Additionally, due to the need to compute multiple hash centers and perform complex distance calculations, the computational cost of PHE is relatively high, especially on large-scale datasets.

**Questions:**

Apart from the issues mentioned in weakness part, there are some additional questions regarding implementation details:
1. The article mentions that to encourage the model to fully capture intra-class diversity, K categories are equally allocated to each class. I would like to know how significant the choice of K is on the model's performance, and if there are experiments demonstrating the impact of different K values on the model.
2. Similarly, when using the masking strategy, how does the variation in the value of θ affect the training and final performance of the model?
3. What about the performance of different splits of old and new class, further analysis is needed regarding the size of different splits as well as whether the correlation of old and new classes essentially influence the final performance etc. Some in-depth insights are expected.

**Limitations:**

As mentioned in the appendix of the manuscript, although the proposed PHE framework outperforms existing methods in terms of performance, further research is needed to improve its accuracy in recognizing unknown categories. Additionally, the PHE method involves multiple hyperparameters that require careful tuning to achieve optimal performance, increasing the complexity of model debugging and optimization. Moreover, the weights of the component loss functions in the total loss function need to be appropriately set to balance the optimization objectives of different components, which may require extensive experiments to find the suitable weight combination.

---

> ### Author Rebuttal · Authors · 2024-08-07
>
> We sincerely thank the reviewer’s recognition of our motivation and experimental results in addressing the hash sensitivity issue. We value the reviewer’s insightful comments and will include these into our final revision.
>
> **Q1: Improving accuracy in recognizing unknown categories.**
>
> 1. **Limited room for improvement due to the unique challenge of OCD.** Compared with GCD/NCD tasks, OCD tasks **do not** use unlabeled data that might include unknown-category data for model training, which leads to a larger challenge for OCD methods to recognize unknown categories. We discussed the limitation in the main paper. Compared to the SOTA method, SMILE, our PHE achieves better accuracy for unknown categories. On the CUB dataset, our method exceeds SMILE by 4.1%/16% on 12/16 bits, respectively. Thus, as recognized by Reviewer ZQiT and dwc8, such an improvement might be non-trivial for OCD.
>
> 2. **A possible solution in future work.** Due to the constraint of training data in OCD setting, we consider introducing additional knowledge from pre-trained Large Language Models (LLMs). Firstly, we can leverage LLMs to establish a bank of category attribute prototypes that are expected to be shared across both known and unknown categories. Then, during the on-the-fly prediction process, we plan to use LLM+VLM to match the attribute prototypes for unknown categories. Finally, by jointly considering the instance and attribute features, we hope that our PHE can generate more accurate predictions.
>
> **Q2: Computational cost for large-scale datasets**.
> We agree that the computational cost of PHE is relatively high; however, the computational cost mainly depends on the number of known categories, which is not directly related to dataset scale. Specifically, each known category has a hash center, thus hash centers are represented by a tensor $\mathbf h$ with a shape [num_class, bit]. Main calculations involve simple two-dimensional matrix multiplications, which include the dot product of features $\mathbf{b}$ with a shape [batch_size, bit] and hash centers, $\mathbf{b} * \mathbf{h}$, as well as the dot product operations in the calculation of Hamming distances $\mathbf{h} * \mathbf{h}^\text{T}$. Empirically, we have verified the analyses in the table below. It shows that although the Food dataset is larger in scale than CUB-200 and Scars-198, it contains only 100 categories. Therefore, the average training time per sample for the Food dataset is significantly lower than that for CUB and Scars.
>
> | Dataset              | CUB#200 | SCars#198 | Food#101 |
> | -------------------------------- | ------- | --------- | -------- |
> | number of training samples       | 1.5k    | 2.0k      | 19.1k    |
> | training time / minute           | 100.22  | 161.37    | 691.39   |
> | training time per sample / second | 4.01    | 4.84      | 2.17     |
>
> **Q3: The impact of different K values on the model**. We regard the "K categories" in the review comment as the number of prototypes per category, $k$. We have detailed the impact of this hyperparameter in Fig. 4 and Sec.3.4 of the main paper. Specifically, $k=1$ results in suboptimal performance, as it doesn't effectively represent the complexity of a category. Using larger $k$ captures the nuances within a category, which is crucial for fine-grained categories. However, when $k>5$, the improvement is minimal, as validated on two datasets. Therefore, K is not sensitive, and we choose a relatively large value of 10 for all datasets to avoid over-tuning this hyperparameter.
>
> **Q4: The impact of $\theta$ in the masking strategy.** We have added experiments on the two datasets and reported results in the table below. The masking strategy helps reduce redundancy when using multiple prototypes, thus facilitating prototype learning. A smaller $\theta$ value is found to be appropriate; within the range of (0, 0.2), the masking strategy can improve accuracy. When $\theta$ exceeds 0.2, suboptimal results occur on both datasets. We did not fine-tune this parameter for each dataset but instead set it at 0.1 across all datasets.
>
> | Dataset| |CUB| | | SCars| |
> | - | -| - | - | - | - | - |
> | value of $\theta$ |All |Old |New|All|Old|New|
> | 0| 35.5| 53.1| 26.6| 30.7| 60.3| 16.7|
> | 0.05| 36.1| 53.6| **27.4** | **31.6** | **63.4** | 16.3|
> | 0.1| **36.4** | **55.8** | 27.0| 31.3| 61.9| **16.8** |
> | 0.15| 36.0     | 54.3     | 26.9     | 31.4     | 62.2     | 16.6     |
> | 0.2| 35.3     | 52.7     | 26.6     | 30.1     | 59.8     | 15.7     |
>
> **Q5: Results of different dataset splits.** Good suggestion! We added experiments using different proportions of old category selection on the CUB and SCars datasets below, with all accuracy reported and code bits=12. Based on the results, when the proportion of selected old categories is 75%/25%, our PHE outperforms SMILE by an average of 5.95%/1.0%. This indicates that our PHE is more capable of modeling the nuanced inter-category relationships in fine-grained OCD, as the number of categories increases. Results across all datasets and more code bits will be supplemented in the revision.
>
> | Method | CUB-25% | CUB-50% | CUB-75% | Scars-25% | Scars-50% | Scars-75% |
> | -------------- | --------------- | --------------- | --------------- | ----------------- | ----------------- | ---------------- |
> | SMILE          | 19.9            | 32.2            | 41.2            | 12.6              | 26.2              | 37.0             |
> | PHE (ours)      | **21.2**        | **36.4**        | **46.5**        | **13.3**          | **31.3**          | **43.6**         |
>
> **Q6: Hyperparameter section.** 1) To avoid over-tuning hyperparameters, we use a fixed set of hyperparameters for all datasets (obtained based on CUB) to report accuracy, without tuning them individually for each dataset. 2) The hyperparameters we use are relatively robust within a certain range. In the loss function, the difference in loss scale determines the scale of the proportions.

---

> ### Comment · Reviewer_Pa9u · 2024-08-13
>
> Authors have partly addressed my concerns regarding model's stability and split variation, I'm willing to raise my score by one.

---

> ### Author Response · Authors · 2024-08-13
>
> Dear Reviewer Pa9u,
>
> We greatly appreciate your satisfaction with our responses! We will include the important discussions mentioned above in the final manuscript and highlight them. We truly appreciate your efforts!

---

### Official Review · Reviewer_6xA4 · 2024-07-13

**Soundness:** 4
**Presentation:** 4
**Contribution:** 2
**Rating:** 6
**Confidence:** 5

**Summary:**

This paper focuses on On-the-fly Category Discovery (OCD), which is to determine novel categories during inference. OCD methods first compute the hash code of the image, this code then becomes a "cluster index", if not matching with existing code, then it is a novel category.
However, the problem with the previous OCD method (SMILE) is that it becomes highly sensitive when the code length increases (difficult to match a new category due to the high possible combinations of hash code).
Instead of contrastive learning in SMILE, this paper uses cross-entropy loss adapted with a minimal hash distance scheme as regularization to improve performance. Further, the paper also adds cross entropy loss in the feature space to avoid information loss when compressing features into hash code, the feature representation used is a prototype-based model (e.g., ProtoPFormer).
Overall, it enhances SMILE by using better loss functions and different representations.

**Strengths:**

originality: the idea itself is clever and simple, propose/reuse some modules to mitigate the problem in SMILE (to reduce the sensitivity problem). the hamming ball-based inference utilizes the benefit of GV bound.

quality: analyzed the proposed method with many experiments.

clarity: the paper is well-written

significance: somehow significant as OCD is more practical in open-world scenario, and may help improve retrieval systems.

the one insight I got from this paper is that using hash code alone is sensitive when code length increases so it cannot be used for cluster index easily, and we need to make the hash code as compact as possible to reduce sensitivity (i.e., the same class must have the same hash code, by guiding the hash code into center, although already shown in multiple deep hashing papers such as CSQ [1], DPN [2], OrthoHash [3], [4]).

[1] Li Yuan et al. Central Similarity Quantization for Efficient Image and Video Retrieval. CVPR 2020.
[2] Lixin Fan et al. Deep Polarized Network for Supervised Learning of Accurate Binary Hashing Codes. IJCAI 2020.
[3] Jiun Tian Hoe et al. One Loss for All: Deep Hashing with a Single Cosine Similarity based Learning Objective. NeurIPS 2021
[4] Liangdao Wang et al. Deep Hashing with Minimal-Distance-Separated Hash Centers. CVPR 2023.

**Weaknesses:**

Comparison to deep hashing based methods are missing. There are many parts of the designed components similar to deep hashing such as the minimal hash distance formulation is adapted directly from [1]. Part of the loss objective is similar to cosine-similarity-based hashing methods like [2]. Another simple baseline is using ProtoPFormer + Deep Hashing method. From this paper, we do not know whether a naive application of deep hashing-based method is sufficient hence we cannot verify the novelty and effectiveness of the proposed method.

[1] Liangdao Wang et al. Deep Hashing with Minimal-Distance-Separated Hash Centers. CVPR 2023.
[2] Hoe et al. One Loss for All: Deep Hashing with a Single Cosine Similarity based Learning Objective. NeurIPS 2021.

**Questions:**

In terms of the idea, I have no question, I believe the idea is simple and clever use of existing components. However, the lack of comparison to deep hashing methods makes me doubt the novelty and effectiveness of the proposed method/modules.

**Limitations:**

The author did mention limitations in the appendix which I also agree with.

---

> ### Author Rebuttal · Authors · 2024-08-07
>
> We sincerely thank the reviewer for acknowledging that our idea is clever and simple, and for the positive comments on the writing and experiments.
>
> **Q1: More baseline with deep hash methods**.
>
> Following the meaningful suggestion, we have conducted additional comparative experiments with various deep hashing methods on CUB and SCars datasets. The results are reported and summarized as follows.
>
> |                   |          | CUB |-12bit|          | CUB|-16bit|          | CUB|-32bit|          | SCars|-12bit|          | SCars|-16bit|          | SCars|-32bit|
> | ----------------- | -------- | --------- | -------- | -------- | --------- | -------- | -------- | --------- | -------- | -------- | ----------- | -------- | -------- | ----------- | -------- | -------- | ----------- | -------- |
> | Methods           | All      | Old       | New      | All      | Old       | New      | All      | Old       | New      | All      | Old         | New      | All      | Old         | New      | All      | Old         | New      |
> | DPN[1]            | 22.2     | 38.0      | 14.2     | 17.6     | 24.1      | 14.4     | 12.5     | 11.1      | 13.2     | 18.8     | 36.1        | 10.5     | 14.9     | 25.3        | 9.8      | 12.8     | 20.4        | 9.1      |
> | OrthoHash[2]      | 30.0     | 49.2      | 20.5     | 25.2     | 42.1      | 16.7     | 13.6     | 24.8      | 8.0      | 19.6     | 37.2        | 11.1     | 15.0     | 24.5        | 10.4     | 13.2     | 20.0        | 9.8      |
> | CSQ[3]            | -        | -         | -        | 25.2     | 35.3      | 20.1     | 26.1     | 45.3      | 16.5     | -        | -           | -        | 25.4     | 54.8        | 11.2     | 23.1     | 44.1        | 13.0     |
> | CSQ-MC  | -  | -| -  | 29.3  | 56.4  | 15.7 | 26.4 | 50.6  | 14.3 | -   | -  | -  | 27.4 | 59.9        | 11.8     | 26.9     | 61.8        | 10.0     |
> | MDSH[4]| 34.3 | **57.6**  | 22.8| 31.9| 52.7| 21.5 | 27.4| 40.8 | 20.7| 28.8 | 60.2  | 13.7 | 27.7  | 53.4 | 15.3| 25.8| 47.8| 15.2 |
> | MDSH+Hamming Ball| 35.1 | 55.0 | 25.3 | 35.2 | 53.0 | 26.2 | 35.5 | 47.8 | **29.3** | 29.8 | 56.1| **17.1** | 29.9 | 57.2 | **16.5** | 29.5     | 56.2 | **16.6** |
> | PHE (ours) | **36.4** | 55.8 | **27.0** | **37.6** | **57.4**  | **27.6** | **38.5** | **59.9**  | 27.8 | **31.3** | **61.9** | 16.8 | **31.8** | **65.4** | 15.6| **31.5** | **64.0** | 15.8|
>
> *"MC" represents manually obtained centers that meet the GV bound. "-" means results were unavailable due to the requirements of the Hadamard matrix.
>
> **Summarization of Experimental Results.**
>
> 1. **Comparison with methods[1-3] that generate suboptimal hash centers for modeling category information.** Unlike these methods[1-3], which are designed for retrieval tasks and mainly focus on instance-level discrimination, our PHE leverages the GV bound to constrain the learning of hash centers for category discovery. This approach allows the models to be flexible enough to preserve the rich category information contained in feature-level prototypes, while also generating discriminable hash category discriminators. As a result, PHE achieves better results.
> 2. **Comparison with [4] that optimizes only hash code to align with pre-calculated hash centers.** Different from [4] that uses fixed hash centers, our hash centers are mapped from the prototypes of each category and then updated by end-to-end optimization, which can alter the relationships between categories learned in the CPG module, yielding better results compared to MDSH. For example, with code bits=12, our PHE surpasses MDSH by an average of 2.3% across two datasets. Besides the difference in hash center design, our Hamming-ball-based inference design effectively mitigates the hash sensitivity issue. For instance, when code bits=16, the MDSH-Hamming ball exceeds MDSH by an average of 2.75% across two datasets.
>
> **Q2: Comparison with OrthoHash [2] in terms of loss function**.
>
> We agree with that both PHE and OrthoHash utilize cosine similarity to constrain hash features, yet there are significant differences in our design motivations and specific implementations:
> 1. **Differences in design motivation**. OrthoHash focuses on instance retrieval and applies constraints in only the low-dimensional hash space, which tends to lose category-specific information (CSI), which is especially crucial for OCD tasks. In contrast, our hash feature constraint loss $L_f$ and prototype learning loss $L_p$ are applied in hash space and feature space, respectively. The motivation is to capture CSI  as rich as possible in high-dimensional feature space. We empirically find that this design can effectively mitigate CSI loss when projecting to hash space.
>
> 2. **Differences in implementation**. **a) Hash center generation**. Our hash centers are mapped from category prototypes, which preserves the relationships between categories learned in the prototype feature space. OrthoHash uses a matrix whose row vectors are mutually orthogonal as hash centers, which might hinder modeling the relationship between similar categories in OCD tasks. **b) Cosine similarity**. We compute cosine similarity between hash features and hash centers without sign quantization. In contrast, OrthoHash calculates cosine similarity between continuous hash features and binary hash centers.  **c) Update of hash center**. Our centers can be optimized during the training process, while the centers in OrthoHash are fixed.
>
> *Our PHE effectively unifies representation learning and category encoding, achieving better OCD accuracy than OrthoHash. As shown in the table of **Q1**, PHE surpasses OrthoHash by an average of 9% when code bits=12.*
>
> [1] Deep Polarized Network for Supervised Learning of Accurate Binary Hashing Codes.
> [2] One Loss for All: Deep Hashing with a Single Cosine Similarity based Learning Objective.
> [3] Central Similarity Quantization for Efficient Image and Video Retrieval.
> [4] Deep Hashing with Minimal-Distance-Separated Hash Centers.

---

> > ### Comment · Reviewer_6xA4 · 2024-08-13
> > **recommendation**
> >
> > thank you for your response. the experiments now look more complete, as long as the author adds them into the final revision, I have no further questions. I am recommending a weak accept (6).

---

> > > ### Author Response · Authors · 2024-08-13
> > >
> > > Dear Reviewer 6xA4,
> > >
> > > We really appreciate your valuable comments! We will add the important discussions mentioned above in the final manuscript and highlight them.
> > >
> > > Thank you once again for your efforts and contributions!

---

### Author Rebuttal · Authors · 2024-08-06

We sincerely thank the ACs and reviewers for their considerable efforts in handling our paper.

We have appropriately addressed all concerns raised by the reviewers. These include providing more baselines with deep hashing methods (Reviewer #6xA4, #dwc8) and prototype learning methods (Reviewer #ZQiT, #dwc8), conducting additional ablation studies on hyper-parameters (Reviewer #Pa9u), explaining our choices of hyper-parameters (Reviewer #Pa9u, #ZQiT), comparing training efficiencies (Reviewer #ZQiT), and offering a clearer explanation of our contribution and motivation (Reviewer #ZQiT, #dwc8).

**Paper strengths acknowledged by reviewers:**

The motivation and idea are clear and simple (Reviewer #6xA4), the method is easy to follow (Reviewer #ZQiT), effectively mitigates the high sensitivity issues present in current On-the-fly Category Discovery models (Reviewer #6xA4, #Pa9u, #ZQiT, #dwc8), and demonstrates good performance (Reviewer #Pa9u, #ZQiT, #dwc8). The paper is well-written and easy to understand (Reviewer #6xA4, #ZQiT).

We expect the ACs and reviewers to fully consider the following factors when making the final decision: (1) A novel prototypical hash encoding framework for On-the-fly Category Discovery, which effectively mitigates the “high sensitivity issues” of current OCD methods and achieves significant performance improvements; (2) Thanks to the interpretable prototype learning, we provide a perspective for explaining the model's behavior when discovering categories. (3) Comprehensive responses to all the reviewers’ comments.

Please let us know if you have any additional questions or concerns. We are happy to provide further clarification.

Authors of Submission #524

---

### Decision · Program_Chairs · 2024-09-25

**Decision:**

Accept (poster)

**Comment:**

This paper addresses the On-the-fly Category Discovery (OCD) task, which involves utilizing existing category knowledge to recognize both known and unknown categories in new data streams in real-time. A novel framework has been proposed to address this task. In general all the reviewers are positive about the contribution and performance of the proposed work. Thus, the recommendation is to accept the paper. The authors are strongly encouraged to address the comments in the camera-ready version.